# A mosaic adeno-associated virus vector as a versatile tool that exhibits high levels of transgene expression and neuron specificity in primate brain

Kei Kimura [1], Yuji Nagai [2], Gaku Hatanaka [3,4], Yang Fang[3,4], Soshi Tanabe[1], Andi Zheng[1], Maki Fujiwara[1], Mayuko Nakano[1], Yukiko Hori [2], Ryosuke F. Takeuchi [3,4], Mikio Inagaki [3,4], Takafumi Minamimoto [2], Ichiro Fujita [3,4], Ken-ichi Inoue [1,5] ✉ & Masahiko Takada [1] ✉

Recent emphasis has been placed on gene transduction mediated through recombinant adeno-associated virus (AAV) vector to manipulate activity of neurons and their circuitry in the primate brain. In the present study, we created a novel vector of which capsid was composed of capsid proteins derived from both of the AAV serotypes 1 and 2 (AAV1 and AAV2). Following the injection into the frontal cortex of macaque monkeys, this mosaic vector, termed AAV2.1 vector, was found to exhibit the excellence in transgene expression (for AAV1 vector) and neuron specificity (for AAV2 vector) simultaneously. To explore its applicability to chemogenetic manipulation and in vivo calcium imaging, the AAV2.1 vector expressing excitatory DREADDs or GCaMP was injected into the striatum or the visual cortex of macaque monkeys, respectively. Our results have defined that such vectors secure intense and stable expression of the target proteins and yield conspicuous modulation and imaging of neuronal activity.

In recent years, much attention has been attracted to manipulating the activity of neurons and their circuitry by gene transduction with viral vectors[1–3]. Recombinant adeno-associated virus (AAV) vector has particularly been utilized for a variety of cutting-edge techniques, such as optogenetics and chemogenetics (for reviews, see refs. 4–6). Indeed, substantial advances have been made in AAV vector-mediated gene transfer into target neuronal populations, and the validity of this strategy has been revealed in the primate brain as well as in the rodent brain[7–13]. It is generally accepted that the AAV has numbers of serotypes, and that individual serotypes show different infectious properties[14,15]. In gene transfer experiments on primate and rodent brains, vectors based on the AAV serotypes 1, 5, and 9 (AAV1, AAV5, and AAV9) have so far frequently been used. These neurotropic AAV vectors exhibit higher levels of transgene expression than other serotype vectors, and, therefore, they may have been expected to exert adequate effects on modulating neuronal activity and animal's behavior[10,16–19]. However, given that such vectors possess considerable infectivity to glial cells as well as to neurons, there seems to be a

[1]Systems Neuroscience Section, Department of Neuroscience, Primate Research Institute, and Center for the Evolutionary Origins of Human Behavior, Kyoto University, Inuyama, Aichi 484-8506, Japan. [2]Department of Functional Brain Imaging, National Institutes for Quantum Science and Technology, Chiba 263-8555, Japan. [3]Laboratory for Cognitive Neuroscience, Graduate School of Frontier Biosciences, Osaka University, 1-4 Yamadaoka, Suita, Osaka 565-0871, Japan. [4]Center for Information and Neural Networks, National Institute of Information and Communications Technology and Osaka University, 1-4 Yamadaoka, Suita, Osaka 565-0871, Japan. [5]PRESTO, Japan Science and Technology Agency, Kawaguchi, Saitama 332-0012, Japan. ✉e-mail: inoue.kenichi.6z@kyoto-u.ac.jp; takada.masahiko.7x@kyoto-u.ac.jp

serious pitfall in that they cause inflammatory responses due to the glial infection[20–23].

In contrast, the AAV serotype 2 (AAV2) displays extremely high neuron specificity and, hence, low tissue damage[24–27]. Currently, recombinant vectors based on AAV2 are widely available for clinical use. Our previous works have demonstrated in macaque monkeys that optogenetic/chemogenetic manipulations mediated through the AAV2 vector successfully modulated oculomotor/cognitive behaviors of the animals as well as related neuron/pathway activities[13,28]. However, these reports are rather exceptional on account of a lower transgene expression capacity of the AAV2 vector compared with other serotype vectors[21,23]. Thus, various serotypes of AAV vectors are being utilized in world-wide laboratories for gene transfer experiments, especially on the primate brain, and many neuroscientists are still seeking for an AAV vector optimal for monitoring and manipulating target neuron/pathway activity. Conceivably, there is usually a trade-off relationship between the transgene expression capacity (i.e., sensitivity) and the neuron-specific infectivity (i.e., specificity). For improving the quality of gene transduction into the primate brain for diverse purposes, it is essential to create a novel AAV vector with marked superiority in both aspects.

Here, we developed an AAV vector in which capsid was composed of capsid proteins derived from both AAV1 and AAV2. This mosaic vector, termed AAV2.1 vector, was designed to exhibit excellence in transgene expression (for the AAV1 vector) and neuron specificity (for the AAV2 vector) simultaneously. By comparing with those of the original AAV2 and AAV1 vectors, we analyzed the gene transduction property of the AAV2.1 vector following injections of these vectors into the frontal cortex of macaque monkeys. We further examined the AAV2.1 vector property for gene transfer in comparison with the properties of AAV5 and AAV9 vectors. In the present study, not only the transgene expression pattern of this mosaic vector, but also its applicability to chemogenetic manipulation and in vivo calcium imaging was explored in the striatum and visual cortex, respectively.

## Results

### Production of mosaic AAV vectors

We produced two types of mosaic AAV vectors with different ratios of AAV1 and AAV2 capsid proteins: AAV2.1-A vector with a combined capsid of 10% AAV1 capsid protein and 90% AAV2 capsid protein, and AAV2.1-B vector with a combined capsid of 50% AAV1 capsid protein and 50% AAV2 capsid protein. According to previous in vitro studies[29–31], we initially made the AAV2.1-B vector. In our preliminary survey, however, we found that a certain degree of glial infectivity remained in this vector. Therefore, we subsequently made the AAV2.1-A vector in which the ratio of AAV1 capsid protein was decreased to a large extent (10%). A total of 14 different kinds of vectors based on the AAV2, AAV1, and their mosaic vectors were produced for three sets of our experiments (Table 1): morphological analysis of gene transduction properties (Monkeys A-D), chemogenetic modulation of neuronal activities (Monkeys E and F), and in vivo calcium imaging of neuronal activities (Monkeys G and H). The production efficiency (stock titer) of each vector is listed in Table 1. In general, AAV2 vectors had a lower efficiency than AAV2.1 and AAV1 vectors.

### Gene transduction properties of mosaic AAV vectors

First, we investigated gene transduction properties of the mosaic AAV2.1-A and AAV2.1-B vectors by comparing with those of the original AAV2 and AAV1 vectors. Two types of promoters, ubiquitous cytomegalovirus (CMV) promoter and neuron-specific synapsin (Syn) promoter, were utilized for evaluating the neuron specificity of the vectors and analyzing their transgene expression levels within neurons, respectively. We injected the following eight distinct kinds of AAV vectors carrying the mKO1 gene into the medial wall of the frontal lobe (involving areas 4, 6, and 9) in two macaque monkeys (Monkeys A

and B in Table 1): the AAV-CMV-mKO1 vectors decorated with four different compositions of capsids (AAV2, AAV2.1-A, AAV2.1-B, and AAV1) and the AAV-Syn-mKO1 vectors with the same virions (Fig. 1a). All vectors inserted with CMV promoter were injected at the titer of $1.5 \times 10^{13}$ genome copies (gc)/ml, while those inserted with Syn promoter were at the titer of $6.0 \times 10^{13}$ gc/ml to make the transgene expression for the AAV2 vector detectable. The tracks of cortical injections were placed at least 4.5 mm apart from each other, and no marked overlapping of adjacent injections was detected, as also evidenced by the distribution gaps of transgene-expressing neurons.

We compared the intensity of transgene expression among these vectors at their cortical injection sites (Fig. 1b). The transgene expression level was first quantified as an averaged value of the red fluorescent protein (RFP) signal within a circle of 2.0-mm diameter and normalized to the expression level for the AAV2-CMV vector which was defined as 1.0 (Fig. 1c). Concerning the vectors inserted with CMV promoter, both the AAV1 and the AAV2.1-B vectors exhibited somewhat higher levels of transgene expression than the AAV2.1-A vector. With respect to the vectors inserted with Syn promoter, the AAV2.1-A vector displayed the highest expression level among these three vectors. Regardless of the promoter type, the transgene expression levels for the AAV2 vectors were lower than the others; the expression level was only less than half or one-fifth of that for the AAV2.1-A vector (Fig. 1c). We then quantified the number of cortical neurons (NeuN-positive; see below) expressing the transgene at the sites of vector injections (Fig. 1d). Our cell counts using stereology revealed that, irrespective of the promoter type, the AAV2.1-A or AAV2 vector yielded the largest or smallest number of RFP-positive neurons, respectively. The numbers of RFP-positive neurons transduced via the AAV2.1-B and AAV1 vectors were in between. Thus, data obtained from the fluorescence intensity analysis of the vectors inserted with Syn promoter corresponded closely to those from the cell-count analysis for both CMV and Syn promoters (compare Fig. 1c with Fig. 1d), indicating that the AAV2.1-A vector produced a higher level of gene transduction into neurons compared with the other vectors.

Subsequently, we examined the neuron specificity of the mosaic vectors in comparison with the original vectors. Fluorescence histochemistry was applied to analyze colocalization of NeuN as a neuronal marker or S100β as an astroglial marker in RFP-positive cortical cells (Fig. 2a; Monkeys A and B in Table 1). Cell counts were done by means of the stereological method, and the ratio of double-labeled cells to the total RFP-positive cells was calculated. Data on the vectors with ubiquitous CMV promoter showed that the neuron specificity of the AAV2.1-A vector was high enough to reach the level of the AAV2 vector, while the AAV2.1-B vector had certain glial infectivity, the extent of which was close to that of the AAV1 vector (Fig. 2b). The vectors with Syn promoter exhibited almost complete levels of neuron-specific gene expression, except that a trace of transgene expression in S100β-positive cells remained for the AAV1 vector (Fig. 2b). Overall, among the four vectors tested here, the AAV2.1-A vector possessed the capability of most efficient and neuron-specific expression of the transgene. It should be noted that given their glial infectivity, the transgene expression levels for the AAV1 and AAV2.1-B vectors with CMV promoter, which were more intense than that for the AAV2.1-A vector, might have represented the signal issued not only from neurons, but also from glial cells.

We further compared the gene transduction properties of the AAV2.1-A vector with those of the AAV5 and AAV9 vectors (see Table 1), both of which have often been used in previous studies on rodent as well as primate brains[16–19,32–34]. The AAV-CMV-mKO1 vectors decorated with four different compositions of capsids (AAV2.1-A, AAV1, AAV5, and AAV9), and the AAV-Syn-mKO1 vectors with the same virions were injected into the medial frontal cortex of two monkeys (Monkeys C and D in Table 1; Fig. 3a, b). As in the experiments described above, the vectors were injected at the titer of $1.5 \times 10^{13}$ gc/ml (for CMV promoter)

**Table 1 | Summary of recombinant AAV vectors for macaque monkeys**

| Animal ID | Serotype | % Capsid | | Promoter | Gene | Stock titer (gc/ml) | Injection titer (gc/ml) |
|---|---|---|---|---|---|---|---|
| | | AAV1 | AAV2 | | | | |
| Monkey A, Monkey B | AAV2 | 0 | 100 | CMV Syn | mKO1 mKO1 | $5.0 \times 10^{13}$ $8.0 \times 10^{13}$ | $1.5 \times 10^{13}$ $6.0 \times 10^{13}$ |
| | AAV2.1-A | 10 | 90 | CMV Syn | mKO1[i] mKO1[ii] | $2.2 \times 10^{14}$ $2.8 \times 10^{14}$ | $1.5 \times 10^{13}$ $6.0 \times 10^{13}$ |
| | AAV2.1-B | 50 | 50 | CMV Syn | mKO1 mKO1 | $9.3 \times 10^{14}$ $1.0 \times 10^{15}$ | $1.5 \times 10^{13}$ $6.0 \times 10^{13}$ |
| | AAV1 | 100 | 0 | CMV Syn | mKO1 mKO1[iii] | $7.7 \times 10^{14}$ $8.4 \times 10^{14}$ | $1.5 \times 10^{13}$ $6.0 \times 10^{13}$ |
| Monkey C, Monkey D | AAV2.1-A | 10 | 90 | CMV Syn | mKO1 mKO1 | s.a. (i) s.a. (ii) | $1.5 \times 10^{13}$ $6.0 \times 10^{13}$ |
| | AAV1 | 100 | 0 | CMV Syn | mKO1 mKO1 | $8.5 \times 10^{14}$ s.a. (iii) | $1.5 \times 10^{13}$ $6.0 \times 10^{13}$ |
| | AAV5 | - | - | CMV Syn | mKO1 mKO1 | $5.0 \times 10^{14}$ $9.5 \times 10^{14}$ | $1.5 \times 10^{13}$ $6.0 \times 10^{13}$ |
| | AAV9 | - | - | CMV Syn | mKO1 mKO1 | $1.4 \times 10^{15}$ $1.3 \times 10^{15}$ | $1.5 \times 10^{13}$ $6.0 \times 10^{13}$ |
| Monkey E | AAV2 | 0 | 100 | Syn | hM3Dq-IRES-AcGFP | $1.8 \times 10^{13}$ | $1.0 \times 10^{13}$ |
| | AAV2 | 0 | 100 | CMV | hM3Dq-IRES-AcGFP | $1.0 \times 10^{13}$ | $1.0 \times 10^{13}$ |
| | AAV2.1-A | 10 | 90 | Syn | hM3Dq-IRES-AcGFP | $5.0 \times 10^{13}$ | $1.0 \times 10^{13}$ $5.0 \times 10^{13}$ |
| Monkey F | AAV2 | 0 | 100 | Syn | hM3Dq | $1.0 \times 10^{14}$ | $2.0 \times 10^{13}$ |
| | AAV2 | 0 | 100 | Syn | hM3Dq-IRES-AcGFP | $2.0 \times 10^{13}$ | $2.0 \times 10^{13}$ |
| | AAV2.1-A | 10 | 90 | Syn | hM3Dq-IRES-AcGFP | $4.0 \times 10^{13}$ | $2.0 \times 10^{13}$ |
| | AAV1 | 100 | 0 | Syn | hM3Dq-IRES-AcGFP | $8.9 \times 10^{13}$ | $2.0 \times 10^{13}$ |
| Monkey G | AAV2.1-A | 10 | 90 | CaMKIIα | GCaMP6s[iv] | $1.2 \times 10^{14}$ | $1.0 \times 10^{14}$ $2.0 \times 10^{13}$ |
| | | | | Syn | | $1.2 \times 10^{14}$ | $1.0 \times 10^{14}$ $2.0 \times 10^{13}$ |
| Monkey H | AAV2.1-A | 10 | 90 | CaMKIIα | GCaMP6s | s.a. (iv) | $4.0 \times 10^{13}$ |

s.a., same as.

or $6.0 \times 10^{13}$ gc/ml (for Syn promoter). To account for the possibility that the infection property of the vectors differs among cortical areas, AAV2.1-A vectors were injected at three loci of the medial frontal cortex, and AAV1, 5, and 9 vectors were injected almost symmetrically on the side contralateral to each of the AAV2.1-A vector injection sites (Fig. 3a). The transgene expression level was quantified in the same fashion as in the above experiments and normalized to the expression level for the AAV5-CMV vector which was defined as 1.0 (Fig. 3c). We found that each AAV2.1-A vector injection yielded similar results on both RFP expression level and RFP-positive neuron number, indicating that the vector infection efficiency was stable, at least within the frontal cortex. Among the vectors inserted with CMV promoter, the AAV9 vector displayed the greatest capacity in both the RFP expression level and the RFP-positive neuron number, followed by the AAV2.1-A and AAV1 vectors (Fig. 3c, d). Among the vectors inserted with Syn promoter, on the other hand, the AAV2.1-A vector exhibited the capacity of gene transduction as high as the AAV9 vector. Regardless of the promoter type, the gene transduction capacity of the AAV5 vector was lower than the others. Data obtained from the fluorescence intensity analysis of the vectors inserted with Syn promoter were in close register with those from the cell-count analysis using both CMV and Syn promoters (compare Fig. 3c with Fig. 3d).

We then analyzed the neuron specificity of the AAV5 and AAV9 vectors in comparison with the AAV2.1-A and AAV1 vectors. Fluorescence histochemistry was performed to examine colocalization of NeuN or S100β in RFP-positive cortical cells (Fig. 4a). Data on the vectors with CMV promoter revealed that not only the AAV1 vector (see also Fig. 2), but also the AAV5 and AAV9 vectors exhibited distinct levels of glial infectivity, whereas the AAV2.1-A vector displayed the highest level of neuron specificity (Fig. 4b). Such high neuron specificity of the AAV2.1-A vector was consistent across the frontal cortical areas tested. The vectors with Syn promoter had almost perfect neuron-specific gene transduction, except that transgene expression in S100β-positive cells slightly remained for the AAV9 vector (Fig. 4b). Thus, the level of transgene expression especially for the AAV9 vector with CMV promoter, which was apparently more intense than that for the AAV2.1-A vector, might have represented the signal emitted from glial cells as well as from neurons.

To validate that the superiority of the AAV2.1-A vector to the other conventional serotypes of AAV vectors in the primate brain might be common to the rodent brain, we investigated its gene transduction properties (i.e., transgene expression and neuron specificity) in the rat cerebral cortex by comparing with those of the AAV1, 2, 5, and 9 vectors. The AAV-CMV-mKO1 vectors decorated with the five corresponding capsids were injected into the motor cortex of 20 rats at the titer of $7.5 \times 10^{12}$ gc/ml (Table 2, Fig. 5a). The transgene expression level was normalized to the expression level for the AAV2-CMV vector which was defined as 1.0. When both the RFP expression level and the RFP-positive neuron number were compared among the five vectors at their injection sites, we found that the AAV2.1-A and AAV9 vectors displayed equivalently high levels of gene transduction (Fig. 5b, c). By contrast, the gene transduction levels for the AAV1, 2, and 5 vectors were significantly lower. Moreover, the AAV2.1-A vector as well as the AAV2 vector showed a significantly higher level of neuron specificity compared with the AAV1, 5, and 9 vectors (Fig. 5d, e).

We finally analyzed the extent of retrograde or anterograde transneuronal labeling of thalamic neurons since several studies[35,36] have reported remote gene transduction via conventional serotypes of

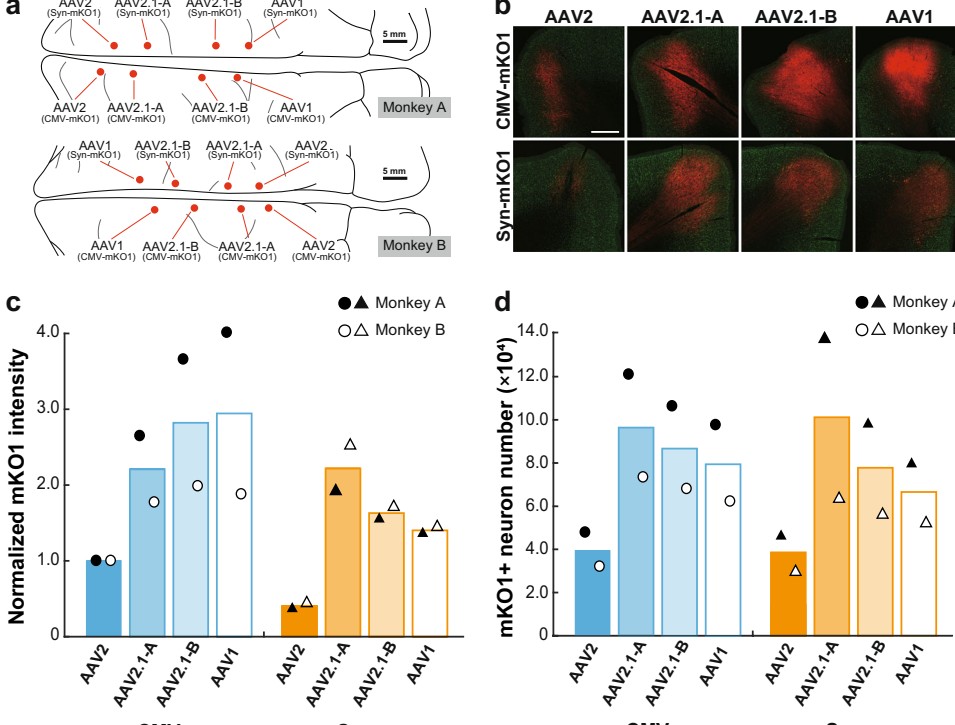

**Fig. 1 | Intensity of mKO1 native-fluorescence and number of mKO1-positive neurons at sites of monkey cortical injections of AAV2, AAV2.1-A, AAV2.1-B, and AAV1 vectors. a** Schematic diagrams showing the injection loci of eight distinct types of recombinant AAV vectors carrying the mKO1 gene in the medial wall of the frontal lobe in Monkey A (upper) and Monkey B (lower). Scale bars, 5 mm. **b** mKO1 native-fluorescence (red) and NeuN immunofluorescence (green) at each vector injection site. Representative images are taken from Monkey A. Similar images were also obtained in Monkey B. Scale bar, 1 mm. **c** Comparison of mKO1 native-fluorescence intensity within a 2-mm-diameter circular observation window. Data are expressed as the averaged value obtained in Monkeys A (filled symbols) and B (open symbols), relative to the mean for the AAV2-CMV vector defined as 1.0. **d** Comparison of mKO1-positive neuron number obtained from stereological cell counts. The cell counts were done over the whole transgene-expressing region in each section. $n = 2$ biologically independent animals. See the Methods for the details of fluorescence intensity and cell-count analyses. Source data are provided as a Source data file.

AAV vectors. In general, the occurrence of remote gene transduction in the thalamus was infrequent (less than 1.0% of the labeled neuron number at the injection site) for the AAV2.1-A vector as well as the AAV2 and 9 vectors (Supplementary Fig. 1a). On the other hand, the AAV1 vector exhibited a significantly higher extent of remote gene transduction into thalamic neurons of the labeled neuron number at the injection site. This implies that the AAV2.1-A vector has a lower remote gene transduction capacity than the AAV1 vector, although it is at least partly composed of the capsid protein of AAV1. It was also found in the monkey that the AAV2.1-A vector injections into the medial frontal cortex resulted in only limited levels of remote gene transduction into thalamic neurons (Supplementary Fig. 1b).

**Application of AAV2.1-A vector to chemogenetic manipulation**
Second, we explored the usefulness of the AAV2.1-A-Syn vector in chemogenetic manipulation of neuronal activity in the primate brain, by applying a recombinant vector carrying the modified human M3 muscarinic receptor (hM3Dq) gene to the excitatory DREADDs (designer receptors exclusively activated by designer drugs) system[37]. The IRES-GFP (green fluorescent protein) sequence was further inserted as a fluorescent protein tag to examine properties of transduced neurons (AAV2.1-A-Syn-hM3Dq-IRES-GFP; Table 1). For comparison, we prepared four different vectors carrying the hM3Dq gene which were based on the original AAV2 or AAV1 vector and inserted with Syn or CMV promoter (AAV2-Syn-hM3Dq-IRES-GFP, AAV2-CMV-hM3Dq-IRES-GFP, AAV2-Syn-hM3Dq, and AAV1-Syn-hM3Dq-IRES-GFP; Table 1). We tested these vectors in terms of the potential of receptor binding, the responsiveness to ligand administration, and the duration of transgene

expression. They were injected into the striatum, the caudate nucleus or putamen, of each hemisphere in two macaque monkeys (Monkeys E and F in Table 1; Figs. 6a, e and 7a, b).

In the first monkey (Monkey E), we compared the AAV2.1-A vector with the AAV2 vector in terms of the three aspects listed above. We injected three types of highly neuron-specific vectors into the striatum: the AAV2.1-A-Syn vector into the caudate nucleus and putamen on one side, and the AAV2-CMV vector into the caudate nucleus and the AAV2-Syn vector into the putamen on the other side (Fig. 6a; see also Fig. 7a). The injection titer of these vectors was set at $1.0 \times 10^{13}$ gc/ml, except that the AAV2.1-A-Syn vector was additionally injected at $5.0 \times 10^{13}$ gc/ml into the putamen since its production efficiency (stock titer) was higher than those of the AAV2-CMV/Syn vectors (Table 1). To evaluate the potential of DREADDs receptor binding in vivo, relevant to the strength of hM3Dq expression, positron emission tomography (PET) imaging with radiolabeled deschlorocloclozapine ([$^{11}$C]DCZ) was carried out (see ref. 38). To monitor the time-dependent changes in hM3Dq expression, we performed [$^{11}$C]DCZ-PET scans in the monkey at four times (30, 45, 80, and 184 days) post-injection (Fig. 6b, c). To verify the responsiveness to DREADDs ligand administration, we next attempted chemogenetic activation of neurons where hM3Dq was expressed through the AAV vectors. PET imaging with radiolabeled fluorodeoxyglucose ([$^{18}$F]FDG) was used for in vivo visualization of glucose metabolism as an index of neuronal/synaptic activation[39–41]. To detect the FDG uptake following systemic administration of DCZ, we performed [$^{18}$F]FDG-PET scans in the monkey twice (86 and 193 days) after the vector injections (Fig. 6d; see also ref. 38).

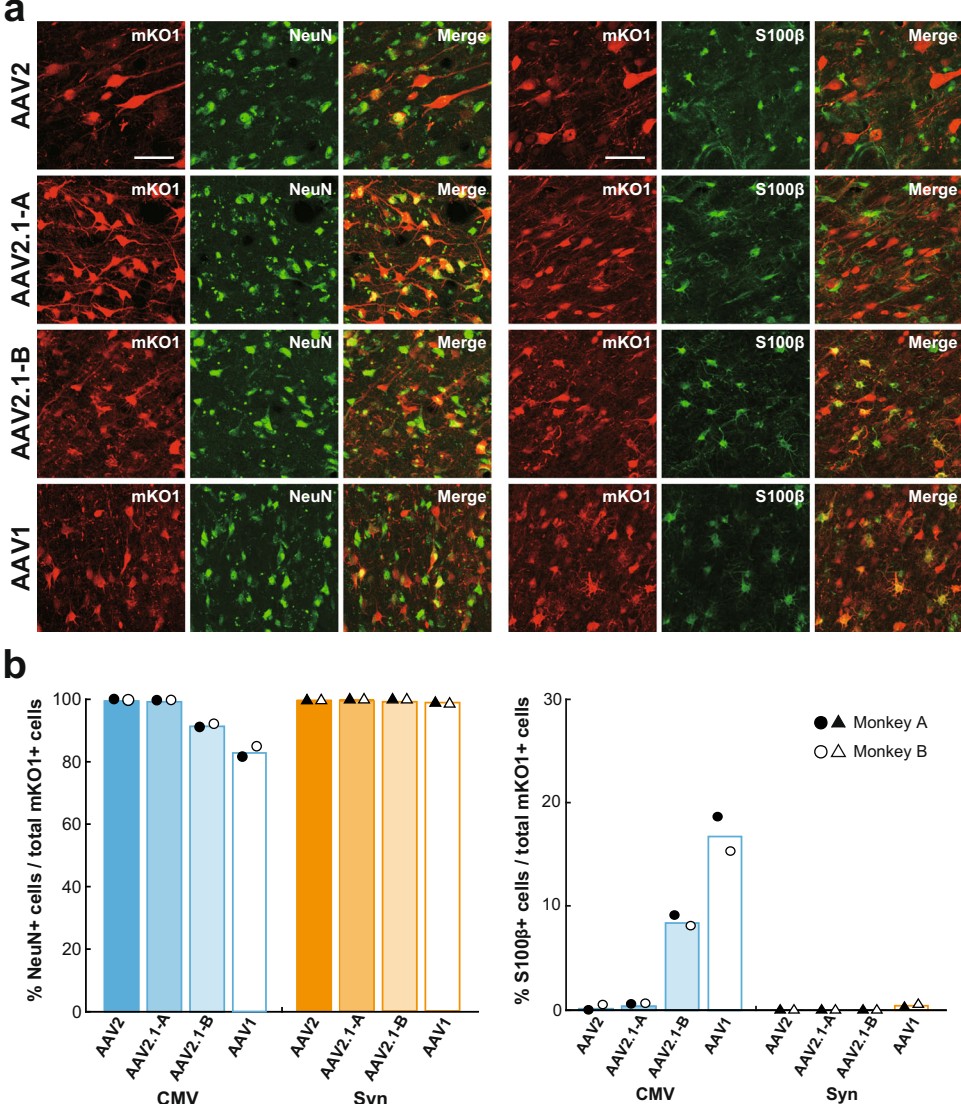

**Fig. 2 | Transgene expression in neurons and glial cells at injection sites of AAV2, AAV2.1-A, AAV2.1-B, and AAV1 vectors. a** Double fluorescence histochemistry for mKO1 native-fluorescence (red) and immunofluorescence for NeuN (green; left) or S100β (green; right) at the injection sites of AAV vectors inserted with CMV promoter. Shown are representative images obtained in Monkey A. In each case, double-labeled cells are denoted in yellow (Merge). Representative images are taken from Monkey A. Similar images were also obtained in Monkey B. Scale bars, 50 μm. **b** Ratios of cells double-labeled for NeuN (left) or S100β (right) to the total mKO1-positive cells at each vector injection site. Each ratio was determined as the mean of data obtained in three sections examined, including the section through the center of the injection site. *n* = 2 biologically independent animals. Source data are provided as a Source data file.

When the vectors were injected at the titer of 1.0 × 10^13 gc/ml, the AAV2.1-A vector exhibited a high level of [^11C]DCZ binding (hM3Dq expression; see ref. 13), whereas the [^11C]DCZ binding levels for the AAV2 vectors were quite low regardless of the promoter type (Fig. 6b, c). In addition, the AAV2.1-A vector injected at a higher titer (5.0 × 10^13 gc/ml) displayed an even greater level of [^11C]DCZ binding compared with the lower-titer one (Fig. 6b, c). The [^11C]DCZ-PET scans showed that the extent of [^11C]DCZ binding for the AAV2.1-A vector was gradually increased until 45 days post-injection and then appeared to reach the plateau. Such hM3Dq expression lasted over six months (Fig. 6b, c). Similar results were obtained from the [^18F] FDG-PET scans. At the injection site of the AAV2.1-A vector at the lower as well as the higher titer, the extent of FDG uptake was markedly increased by DCZ administration at 86 days after the vector injection and retained even at 193 days post-injection (Fig. 6d). By contrast, virtually no increase was found for either type of the AAV2 vectors (Fig. 6d).

In the second monkey (Monkey F), we injected four types of vectors into the striatum: the AAV2-Syn vector into the caudate nucleus and the AAV2.1-A-Syn vector into the putamen on one side, and the AAV2-Syn vector without GFP tag into the caudate nucleus and the AAV1-Syn vector into the putamen on the opposite side (Fig. 6e; see also Fig. 7b). The injection titer of these vectors was set at 2.0 × 10^13 gc/ml. Since the AAV2 vector without fluorescent protein tag and the AAV1 vector have been used in previous DREADDs studies in which animal behavior was successfully manipulated[13,38], the validity of the AAV2.1-A vector was analyzed in comparison with these conventional vectors. In this monkey, [^11C]DCZ-PET scans were carried out at seven times (30, 45, 64, 120, 143, 176, and 358 days) after the vector injections. Similar high levels of [^11C]DCZ binding (hM3Dq expression) were observed for all of the AAV2 vector without GFP tag, the AAV1 vector, and the AAV2.1-A vector (Fig. 6f, g). The binding levels were gradually increased until 64 days post-injection and then reached the plateau. Such hM3Dq expression was retained

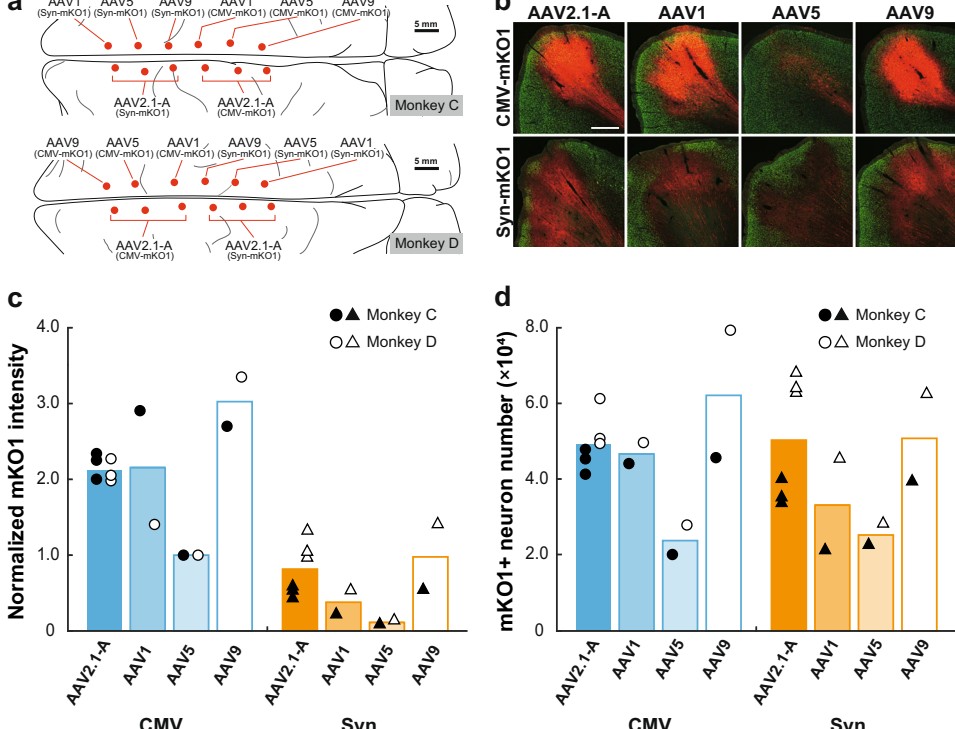

**Fig. 3 | Intensity of mKO1 native-fluorescence and number of mKO1-positive neurons at sites of monkey cortical injections of AAV2.1-A, AAV1, AAV5, and AAV9 vectors. a** Schematic diagrams showing the injection loci of eight distinct types of recombinant AAV vectors carrying the mKO1 gene in the medial wall of the frontal lobe in Monkey C (upper) and Monkey D (lower). The sites of cortical injections of the AAV2.1-A vector were placed symmetrically to those of the other vectors in the opposite hemisphere. Scale bars, 5 mm. **b** mKO1 native-fluorescence (red) and NeuN immunofluorescence (green) at each vector injection site. Representative images are taken from Monkey C. Similar images were also obtained in

Monkey D. Scale bar, 1 mm. **c** Comparison of mKO1 native-fluorescence intensity within a 2-mm-diameter circular observation window. Data are expressed as the averaged value obtained in Monkeys C (filled symbols) and D (open symbols), relative to the mean for the AAV5-CMV vector defined as 1.0. **d** Comparison of mKO1-positive neuron number obtained from stereological cell counts. The cell counts were done over the whole transgene-expressing region in each section. See the Methods for the details of fluorescence intensity and cell-count analyses. $n = 2$ biologically independent animals. Source data are provided as a Source data file.

as long as one year (Fig. 6f, g). With respect to the FDG uptake, we performed [¹⁸F]FDG-PET scans five times (49, 72, 129, 182, and 365 days) after the vector injections and observed that the AAV2.1-A vector, together with the AAV2 vector without GFP tag and the AAV1 vector, displayed a high level of FDG uptake induced by DCZ administration (Fig. 6h). Such responsiveness of striatal neurons to the DREADDs ligand also lasted over one year. The FDG uptake for the AAV2 vector without GFP tag seemed higher than the AAV2.1-A and AAV1 vectors. Consistent with the findings obtained in the first monkey, the AAV2-Syn vector with GFP tag exhibited much lower levels of [¹¹C]DCZ binding and FDG uptake than the other three vectors (Fig. 6h).

We histologically confirmed the PET imaging data described above. Our immunohistochemical analysis with anti-GFP or anti-muscarinic acetylcholine receptor M3 (M3) antibody demonstrated that the sites of hM3Dq expression visualized by [¹¹C]DCZ-PET imaging corresponded to GFP-positive regions in the striatum (Fig. 7a, b). To verify neuronal activation chemogenetically induced by DCZ administration 2 hr before sacrifice, the density of striatal neurons expressing c-fos was examined at the injection site of each vector (Fig. 7c, d). Consistent with the findings obtained from [¹⁸F]FDG-PET imaging, a large number of c-fos-expressing neurons were found after DCZ administration at the striatal injection sites of the AAV2.1-A vector including the higher-titer one, as compared to the AAV2 vector with GFP tag (Fig. 7c–f). On the other hand, striatal neurons expressing c-fos were more frequently seen at the injection sites of the AAV2 vector without the GFP tag and the AAV1 vector (Fig. 7d, f). Our GFP

immunohistochemical observations revealed that axon terminal labeling from the striatal injection site was evident in the substantia nigra (pars reticulata), whereas only very rarely was found cell body labeling in the substantia nigra (pars compacta), thalamus, or cerebral cortex, all of which are known to send projection fibers to the striatum (Fig. 7g). This indicates that the AAV2.1-A vector was preferentially taken up from cell bodies and dendrites, but not from axon terminals.

We characterized striatal neurons in which the transgene was expressed via the AAV2.1-A vector by identifying their neuron types (i.e., projection neurons vs. interneurons). Fluorescence histochemistry was performed to analyze colocalization of dopamine- and cAMP-regulated phosphoprotein (DARPP-32) as a medium spiny projection neuron marker, or parvalbumin (PV)/choline acetyltransferase (ChAT) as fast-spiking/cholinergic interneuron markers in GFP-positive cells. We found that the AAV2.1-A vector was transduced into both projection neurons and interneurons in the striatum (Supplementary Fig. 2).

**Application of AAV2.1-A vector to in vivo calcium imaging**
Third, we investigated applicability of the AAV2.1-A vector to in vivo calcium imaging of neuronal activities in the primate cerebral cortex. We produced a recombinant vector expressing GCaMP6s and injected it into the primary and secondary visual cortical areas (V1 and V2) of two macaque monkeys (Monkeys G and H in Table 1).

We first performed intrinsic signal optical imaging to visualize orientation maps of V1 and V2 by examining response signals to drifting gratings of various orientations. We then carried out one-photon wide-field calcium imaging and two-photon calcium

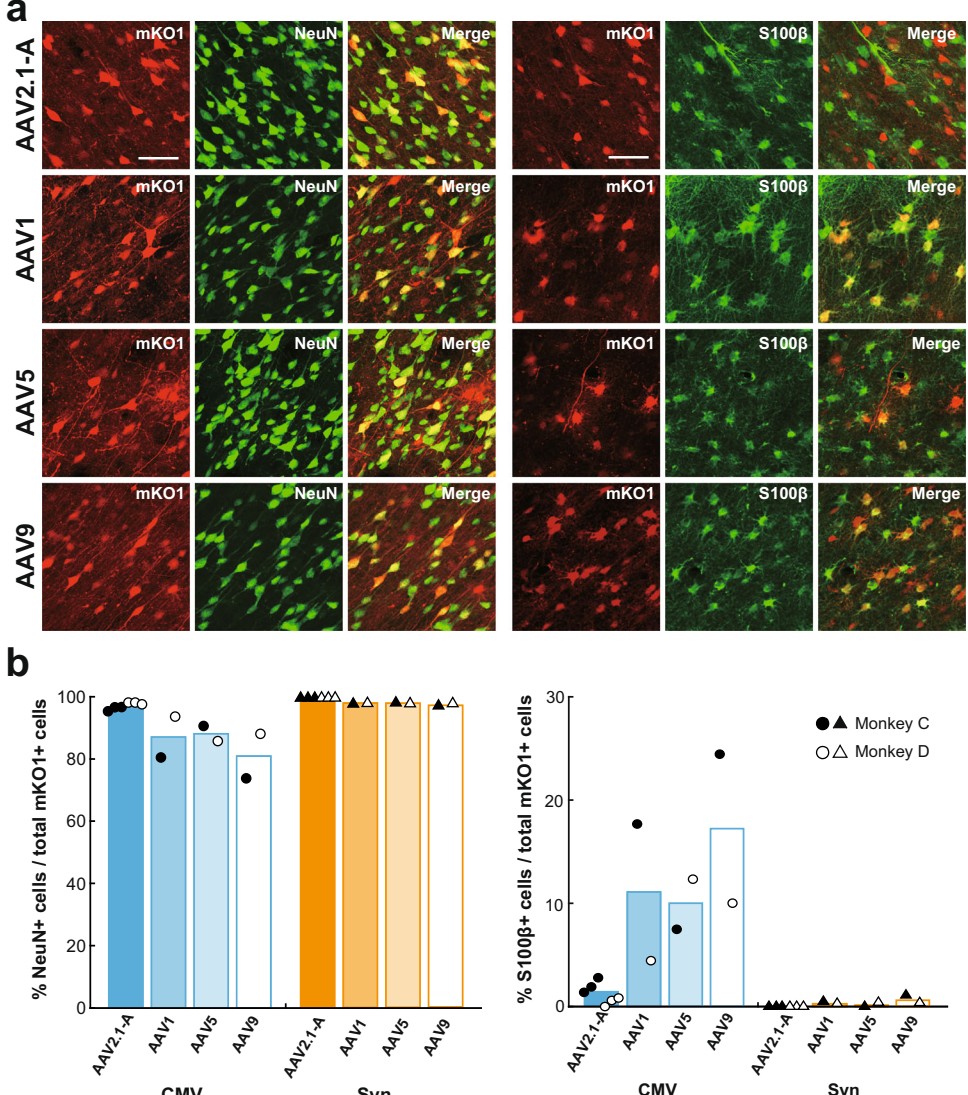

**Fig. 4 | Transgene expression in neurons and glial cells at injection sites of AAV2.1-A, AAV1, AAV5, and AAV9 vectors. a** Double fluorescence histochemistry for mKO1 native-fluorescence (red) and immunofluorescence for NeuN (green; left) or S100β (green; right) at the injection sites of AAV vectors inserted with CMV promoter. Shown are representative images obtained in Monkey C. In each case, double-labeled cells are denoted in yellow (Merge). Representative images are taken from Monkey C. Similar images were also obtained in Monkey D. Scale bars, 50 μm. **b** Ratios of cells double-labeled for NeuN (left) or S100β (right) to the total mKO1-positive cells at each vector injection site. Each ratio was determined as the mean of data obtained in three sections examined, including the section through the center of the injection site. *n* = 2 biologically independent animals, and *n* = 6 independent injections for the AAV2.1-A vector. Source data are provided as a Source data file.

## Table 2 | Summary of recombinant AAV vectors for rats

| Animal ID | Serotype | % Capsid | | Promoter | Gene | Stock titer (gc/ml) | Injection titer (gc/ml) |
|---|---|---|---|---|---|---|---|
| | | AAV1 | AAV2 | | | | |
| #01-04 | AAV2.1-A | 10 | 90 | CMV | mKO1 | $5.0 \times 10^{13}$ | $7.5 \times 10^{12}$ |
| #05-08 | AAV1 | 100 | 0 | CMV | mKO1 | $8.5 \times 10^{14}$ | $7.5 \times 10^{12}$ |
| #09-12 | AAV2 | 0 | 100 | CMV | mKO1 | $5.0 \times 10^{13}$ | $7.5 \times 10^{12}$ |
| #13-16 | AAV5 | - | - | CMV | mKO1 | $5.0 \times 10^{14}$ | $7.5 \times 10^{12}$ |
| #17-20 | AAV9 | - | - | CMV | mKO1 | $1.4 \times 10^{15}$ | $7.5 \times 10^{12}$ |

imaging to capture fluorescence signals from neurons in V1 (Fig. 8a for Monkey H; Supplementary Fig. 3 for Monkey G. In both monkeys, V1 neurons loaded with the recombinant DNA exhibited fluorescence responses time-locked to visual stimulation. The responses of were pronounced, exhibiting Δ*F/F* as high as 400% in some neurons (Fig. 8b, c). They were selective to a particular range of orientation of gratings (Fig. 8c). The orientation selectivity of imaged neurons was consistent with that determined by intrinsic optical signal recording and by one-photon wide-field calcium imaging (Fig. 8a). Repeated imaging

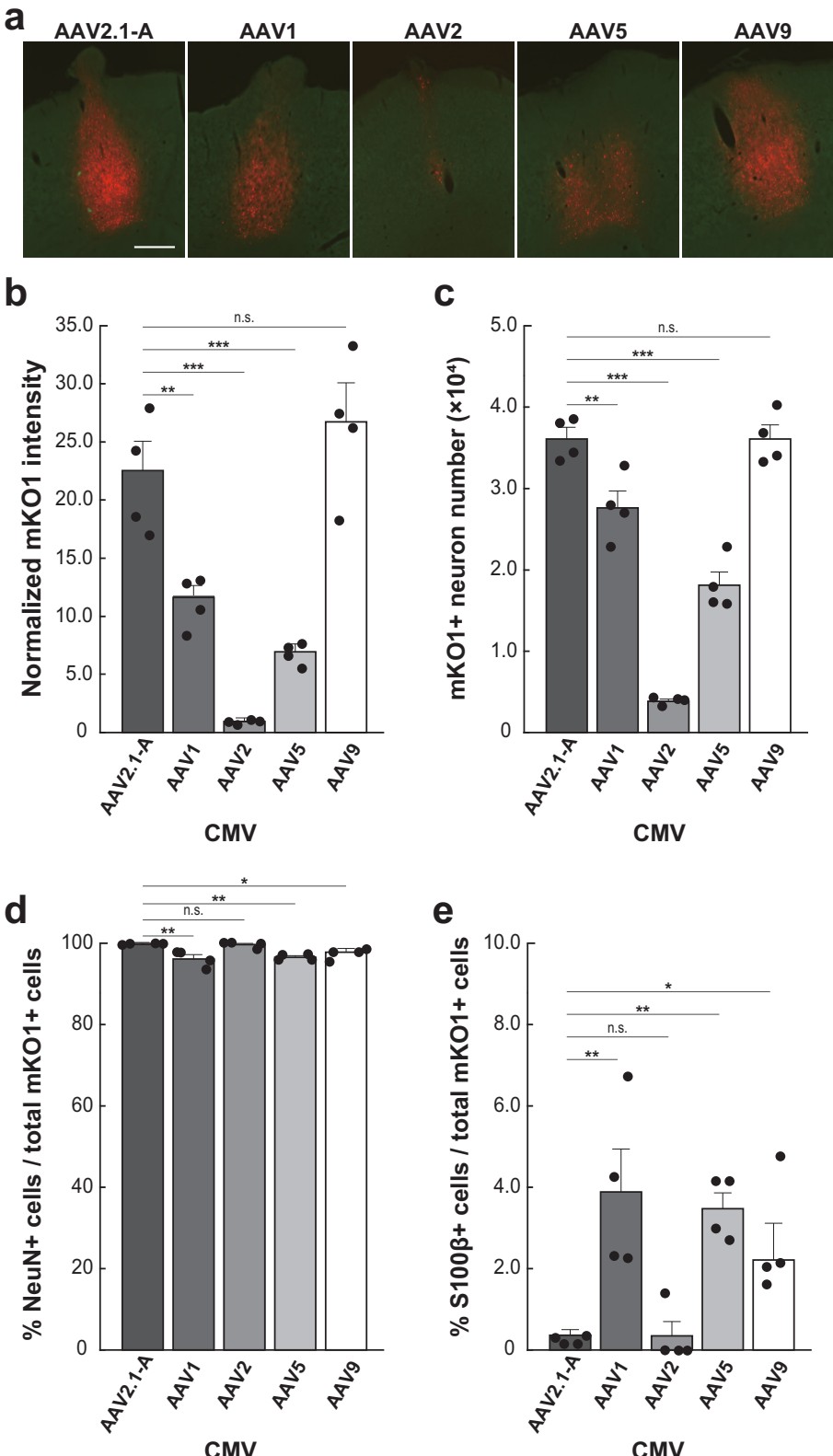

showed that the fluorescence signals could be recorded over eight months after the vector injection. Even at 244 days post injection, the captured signals were comparable to those at 41 days post-injection in Monkey H (Fig. 8b). Intrinsic signal optical imaging revealed stable orientation maps without any unresponsive local regions over this prolonged experimental period, indicating that the viral injection did not cause noticeable damage in the cortex. The results demonstrate that the mosaic vector we developed provides a sensitive, reliable, and stable tool to monitor neuronal activities in the primate cerebral cortex.

## Discussion
In the present study, we have newly developed the mosaic AAV vector, termed AAV2.1 vector, of which capsid was composed of capsid

**Fig. 5 | Gene transduction properties of AAV2.1-A, AAV1, AAV2, AAV5, and AAV9 vectors at their injection sites in rat cerebral cortex. a** mKO1 native-fluorescence (red) and NeuN immunofluorescence (green) at each vector injection site. Images are representative of *n* = 4 animals per group. Scale bar, 500 μm. **b** Comparison of mKO1 native-fluorescence intensity within a 1-mm-diameter circular observation window at the injection sites of the AAV2.1-A, AAV1, AAV2, AAV5, and AAV9 vectors. Data are expressed as the averaged value obtained in four rats, relative to the mean for the AAV2-CMV vector defined as 1.0. Each dot represents the intensity value for each individual. **c** Comparison of mKO1-positive neuron number obtained from stereological cell counts. The cell counts were done over the whole transgene-expressing region in each section. See the Methods for the details

of fluorescence intensity and cell-count analyses. Each dot represents the neuron number for each individual. **d** Ratios of cells double-labeled for NeuN to the total mKO1-positive cells at each vector injection site. Each ratio was determined as the mean of data obtained in three sections examined, including the section through the center of the injection site. Each dot represents the ratio for each individual. **e** Ratios of cells double-labeled for S100β to the total mKO1-positive cells. Each ratio was determined as above. Each dot represents the ratio for each individual. Significant difference from the AAV2.1 vector (one-way ANOVA followed by post hoc Dunnett's test): *$P < 0.05$, **$P < 0.01$, ***$P < 0.001$, n.s. $P > 0.05$. Source data are provided with this paper. The exact $P$ values are also available as a Source Data file. Error bars, SEM. $n = 4$ biologically independent animals.

proteins derived from both AAV1 and AAV2. Of the two types of mosaic vectors with different compositions of their capsid proteins, the AAV2.1-A vector of which capsid was obtained from a higher ratio of AAV2 capsid protein (10% AAV1 and 90% AAV2) has successfully met the two requirements, high levels of transgene expression and neuron specificity. Our analysis of the monkey cerebral cortex has revealed that this vector possesses excellence in transgene expression (for the AAV1 vector) and neuron specificity (for the AAV2 vector) simultaneously (see Figs. 1 and 2). It should be noted that similar AAV vectors, decorated with a mosaic capsid consisting of AAV1 and AAV2 capsid proteins, showed enhanced levels of transgene expression in the muscle and liver in mice[29]. In this previous study[29], a mosaic vector obtained from the same ratio of both capsid proteins, equivalent to our AAV2.1-B vector, has the advantage of gene transduction over those from their different ratios. Although such a discrepancy may be ascribable to the differences in target organs and/or animal species, the composition of capsid proteins is certainly responsible for the gene transduction properties of the vectors. Among the vectors inserted with CMV promoter, the intensity of transgene expression for the AAV2.1-A vector was lower than those for the AAV1 vector and the AAV2.1-B vector. When the CMV promoter was replaced with the Syn promoter, however, the AAV2.1-A vector exhibited the highest level of transgene expression. In view of the fact that the AAV1 and AAV2.1-B vectors with CMV promoter displayed some extent of glial infection, the intensity of their transgene expression might have represented signals derived from both neurons and glial cells. Conversely, transgene expression in neurons via these vectors was limited under the control of Syn promoter. The data obtained from the fluorescence intensity analysis of the vectors with Syn promoter corresponded closely to those from the cell-count analysis using both CMV and Syn promoters. This implies that differential transgene expression levels for the vectors inserted with Syn promoter reflect the difference in the number of transgene-expressing neurons, albeit distinct expression levels within individual neurons cannot as yet be excluded.

Although the transgene expression efficiency and neuron specificity of the AAV2.1 vector were stable in the medial frontal cortex of the monkey, it should be noted here that this result was obtained from the comparison of many serotypes of AAV vectors injected into distinct cortical areas of four monkeys. In the monkey experiments, it is quite difficult to minimize a bias caused by the individual differences, unlike the rodent experiments which are easy to make subjects' properties uniform, such as genetic background, sex, age and weight. Therefore, it is not so realistic to compare statistically the properties of numbers of vectors in the monkey brain under the same conditions. In our study, the highest number of neurons were transduced at the site of each vector injection in Monkey A whose age was 18 years old. On the other hand, the number of transduced neurons was fewer and similar in Monkeys B-D who were relatively close in age (Monkey B, 11 years old; Monkey C, 8 years old; Monkey D, 12 years old). Although the age of subjects may have affected the transgene expression efficiency or neuron specificity of the AAV vectors, the present results were stable when the levels of transgene expression were compared in the

AAV2.1-A vector and the other serotype vectors which were injected in single monkeys. Furthermore, we found that higher neuron specificity (i.e., lower glial infectivity) of the AAV2.1-A vector than that of the AAV1, 5, and 9 vectors were stable among the monkeys.

In the present study, we have extended our histological analyses of gene transduction properties of the AAV2.1-A vector to the rat cerebral cortex by comparing with those of the AAV1, 2, 5, and 9 vectors (see Fig. 5). We found that the AAV2.1-A vector exhibited a significantly higher number of transgene-expressing neurons than the AAV1, 2, and 5 vectors, and a significantly lower rate of glial infectivity than the AAV1, 5, and 9 vectors. These findings were largely consistent with those obtained in the monkey cerebral cortex, thus indicating the utility and versatility of this mosaic vector across the animal species. In the present study, however, we have only reported the results on the comparison between the properties of the AAV2.1-A vector and the other conventional serotypes of AAV vectors in the frontal cortex. As a future study, an attempt should be made to show the transgene expression patterns via these vectors following their injections into other brain regions, for instance, the basal ganglia.

Moreover, the AAV2.1-A vectors expressing excitatory DREADDs and GCaMP have successfully been applied to chemogenetic manipulation and in vivo calcium imaging, respectively, in the primate brain (see Figs. 6–8, Supplementary Fig. 3). These AAV2.1-A vectors secured intense and stable expression of the target protein to achieve unequivocal modulation and imaging of neuronal activities. The overall data in our chemogenetic manipulation experiments have demonstrated that the AAV2.1-A vector displayed high levels of both the potential of DREADDs receptor binding and the responsiveness to DREADDs ligand administration. It should also be mentioned that the AAV2.1-A vector employed at a higher titer yielded a greater capacity of transgene expression (see Table 1). The transgene expression capacity of the AAV2 vector was lower regardless of the promoter type even compared with the AAV2.1-A vector at the same titer. In striking contrast, the capacity of the AAV2 vector without a fluorescent protein tag became much higher, especially in terms of the responsiveness to DREADDs ligand administration. It is generally accepted that insertion of the IRES-GFP sequence into conventional AAV vectors dampens transgene expression[42–44]. However, given its potentially high level of transgene expression, the AAV2.1-A vector can be considered to retain a sufficient expression level even with the fluorescent protein tag inserted. While other non-fluorescent tag proteins are available, AAV vectors expressing no marker proteins within either somas or nuclei do not likely merit application to anatomical studies which require histological determination of the localization, number, and/or type of transduced cells.

The AAV2.1-A has accomplished both the high-level target gene expression in neuronal cells and the minimal infection to glial cells. To achieve the high-yield transgene expression in the primate brain, the AAV1, AAV5, and AAV9 vectors have often been utilized[10,16–19,23]. In the present study, the glial, as well as neuronal infectivity, was detected in

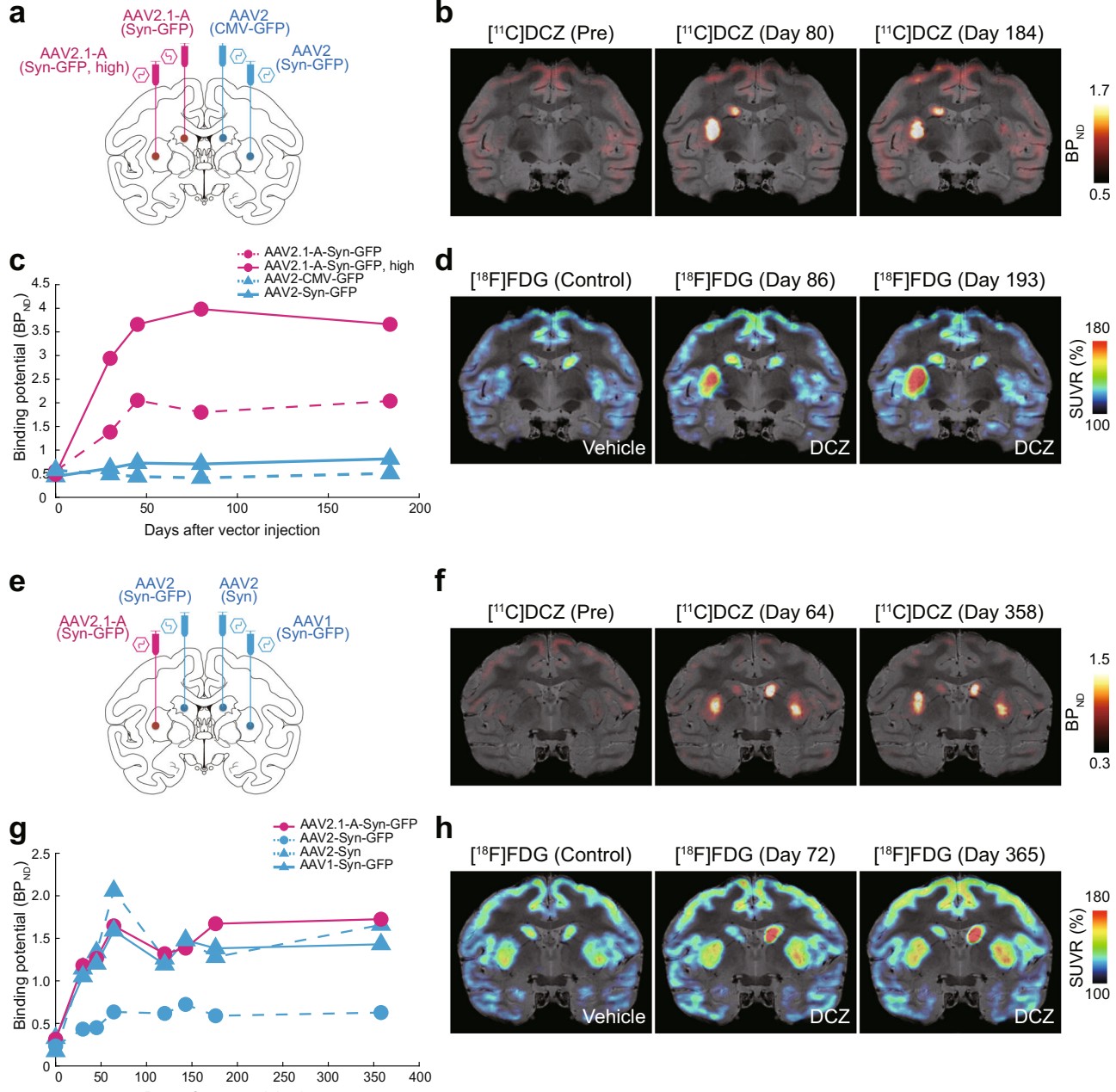

**Fig. 6 | Chemogenetic manipulation of striatal neuron activity. a** Schematics showing the sites of vector injections in the striatum in Monkey E. AAV2.1-A-Syn-GFP vector into the caudate nucleus and the same vector at a higher-titer (Syn-GFP, high) into the putamen on one side, and AAV2-CMV-GFP vector into the caudate nucleus and AAV2-Syn-GFP vector into the putamen on the other side. **b** Representative PET images overlaid on MR images showing [11C]DCZ-specific binding taken before (Pre) or at 80 and 184 days after the vector injections. Each value of binding potential is expressed as the regional binding potential relative to a nondisplaceable radioligand in tissue (BP$_{ND}$). **c** Time-dependent changes of [11C] DCZ-specific binding at the sites of vector injections. Values before the injections, corresponding to (Pre) in **b**, are plotted at Day 0. **d** Representative PET&MR-fused images showing normalized [18F]FDG uptake following vehicle administration (Control) or DCZ administration at 86 and 193 days after the vector injections. Each value of FDG uptake is expressed as the standardized uptake value ratio (SUVR) to

the mean value of the whole brain, averaged between 60 and 90 min after the radioligand injection. **e** Schematics showing the sites of vector injections in Monkey F. AAV2-Syn-GFP vector into the caudate nucleus and AAV2.1-A-Syn-GFP vector into the putamen on one side, and AAV2 vector with Syn promoter but without GFP tag (AAV2-Syn) into the caudate nucleus and AAV1-Syn-GFP vector into the putamen on the other side. **f** Representative PET images overlaid on MR images showing [11C] DCZ-specific binding taken before (Pre) or at 64 and 358 days after the vector injections. **g** Time-dependent changes of [11C]DCZ-specific binding at the sites of vector injections. Values before the injections are plotted at Day 0. **h** Representative PET&MR-fused images showing normalized [18F]FDG uptake following vehicle administration or DCZ administration at 72 and 365 days after the vector injections. Source data are provided in this paper. Schematics in **a** and **e** were reproduced from Paxinos G. et al. [57]. (This article was published in Academic Press, Copyright Elsevier (2009)).

the AAV1, 5, and 9 vectors (see Figs. 3 and 4). For the AAV1, 5, and 9 vectors to secure neuron-specific gene expression, the selection of a promoter, like Syn promoter, is therefore critical. However, even though the Syn promoter restricts the cell type to neurons in which the

transgene is expressed via the vectors, it is still impossible to avoid infection to non-neuronal cells (i.e., glial cells) with these serotypes per se, and transgene expression due to leaky Syn promoter activity or ITR promoter activity may occur as we observed in the case of AAV1-

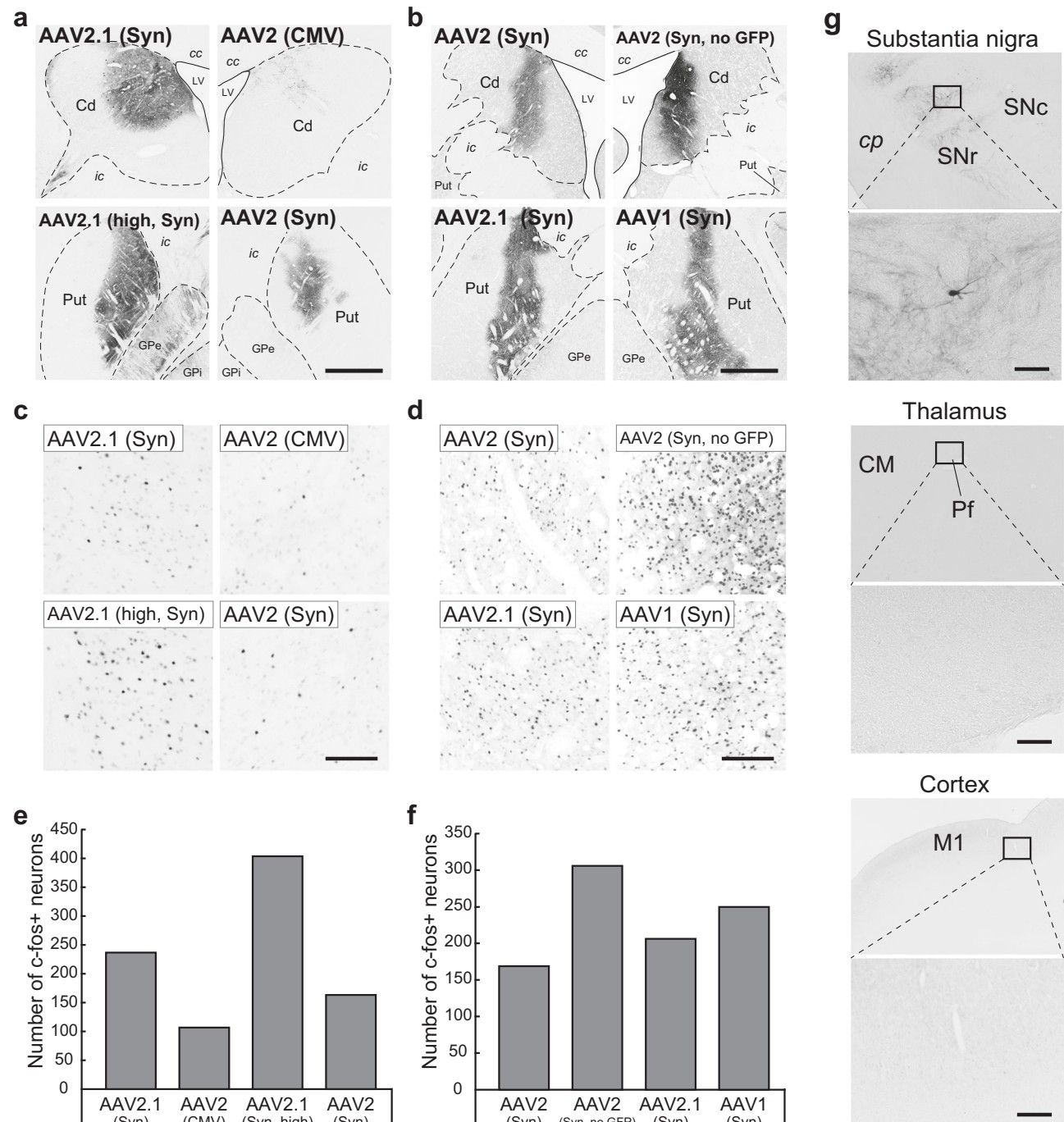

**Fig. 7 | Histological analyses of striatal sections subjected to chemogenetic manipulation. a**, **b** Sites of vector injections immunostained with anti-GFP (**a**) or anti-M3 (**b**) antibody in Monkeys E and F, respectively. Coronal sections. Scale bar, 1 mm. cc corpus callosum, Cd caudate nucleus, GPe external segment of the globus pallidus, GPi internal segment of the globus pallidus, ic internal capsule, LV lateral ventricle, Put putamen. **c**, **d** Immunostaining for c-fos at individual vector injection sites corresponding to **a**, **b**. Scale bar, 100 μm. **e**, **f** Number of c-fos-positive neurons within a 1-mm-diameter circular observation window at individual vector injection sites corresponding to **a**, **b**. Expressed as the mean value of cell counts obtained from three equidistant sections. The cell counts were done over the whole transgene-expressing region within the Cd or Put in each section. The averaged number of c-fos positive neurons at the densest area is chosen as a representative value. See the Methods for the details of the cell-count analysis. **g** GFP immunostaining showing anterograde labeling in the substantia nigra pars reticulata (SNr, upper) and thalamus (lower) on the side ipsilateral to the AAV2.1-A vector injection. Anterograde axonal labeling is also seen in the GPe and GPi in the lower-left panel of **a**. Only a single neuron is retrogradely labeled in the substantia nigra pars compacta (SNc). Data were obtained in Monkey E. CM centromedian thalamic nucleus, cp cerebral peduncle, Pf parafascicular thalamic nucleus. Scale bars, 100 μm. Source data are provided as a Source data file.

and AAV9-Syn-mKO1 vector injections into the monkey cortex[45]. Such viral infection to and transgene expression in glial cells may cause inflammatory changes in the brain. Indeed, it has been reported that the conventional serotypes of AAV vectors, especially the AAV9 vector, induce inflammatory responses due to glial transduction[22,23,46–50].

Presumable tissue damage caused by the inflammatory responses discourages their application to long-term experiments over the years in primates. Thus, the AAV2.1-A permits behavioral and electrophysiological monitoring over a long time period by a potential reduction in inflammatory responses.

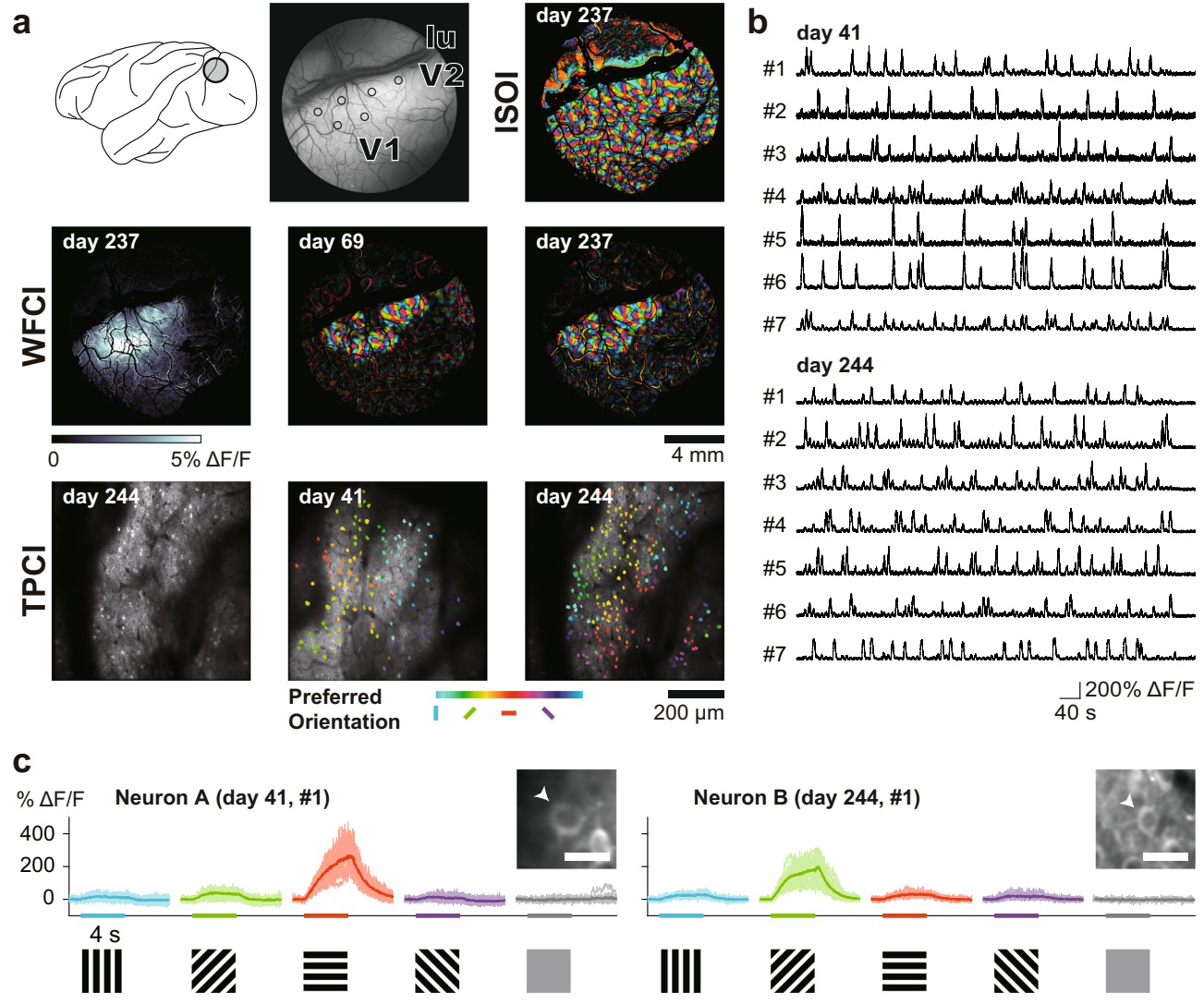

**Fig. 8 | In vivo calcium imaging of visual cortical neuron activity. a** Upper row: left, a recording chamber placed over the V1 and V2 in Monkey H; Central, six loci of AAV2.1-A vector (AAV2.1-A-CaMKIIα-GCaMP6s) injections (denoted by open circles). lu, lunate sulcus; Right, intrinsic signal optical imaging (ISOI) for visualizing an orientation map of the visual cortex at 237 days post-injection. Middle row: One-photon wide-field calcium imaging (WFCI). Left, vector injection sites at 237 days post-injection visualized by selecting the maximum fluorescence changes (ΔF/F) for each pixel across all stimulus conditions; Central & Right, orientation maps visualized at 69- and 237 days post-injection. The color code indicates the preferred orientation determined for each pixel. Note that these maps are basically consistent with the one obtained from ISOI (see the upper right) and, also, at the two-time points apart ~6 months. Lower row: Two-photon calcium imaging (TPCI). Left, structural image of the recording sites at 244 days post-injection obtained by averaging the fluorescence across all frames for a recording session. Central & Right, orientation maps at the single-neuron resolution at 41- and 244 days post-injection. The color code represents the preferred orientation determined for each neuron. **b** Time-dependent changes (ΔF/F) in fluorescent signals obtained from 14 exemplified neurons in the V1 recorded at 41 and 244 days after the vector injection. These calcium transients were recorded in response to visual stimuli. **c** Responses of fluorescent signals of two V1 neurons (Neurons A and B) to drifting gratings of different orientations. The responses to the same orientation with opposite moving directions are superimposed. (Insets) Arrowheads point to the imaged neurons with the typical nucleus-empty appearance of GCaMP labeling. Data were obtained in Monkey H. Scale bars, 20 μm.

For recent years, many neuroscientists have made an effort to achieve reliable and stable modulation of neuronal activity and animal behavior by combining the viral vector system with various cutting-edge techniques. The AAV2.1-A vector developed in the present study bears a prominent advantage over the conventional AAV vectors in view of both transgene expression and neuron specificity in the primate brain. Such a mosaic AAV vector can be a versatile tool for diverse in vivo approaches, including chemogenetic manipulation and in vivo calcium imaging as we tested here. The AAV2.1-A vector exhibited excellent neurotropism even without Syn promoter. This enables us to replace the Syn promoter with other types of promoters so as to modify the transgene expression pattern or achieve the neuron type-selective gene transduction. The widespread and effective application of the AAV2.1-A vector not only could facilitate elucidating neural network mechanisms underlying various brain functions, but also might subserve the development valid primate models of neurological/psychiatric disorders.

## Methods

### Animals

Eight adult macaque monkeys were used for three sets of our experiments (Monkeys A-H in Table 1): two rhesus monkeys (*Macaca mulatta*), 8 and 18 years old, females, 5.6 and 7.8 kg; one cynomolgus monkey (*Macaca fascicularis*), 4 years old, male, 5.2 kg; five Japanese monkeys (*Macaca fuscata*), 3–12 years old, one male and four females, 5.3–8.3 kg. Twenty Wister rats (12–13 weeks old, males, 260–318 g) were also used.

For the first set of our experiments with Monkeys A-D and the rats on gene transduction properties of mosaic AAV vectors, the experimental protocol was approved by the Animal Welfare and Animal Care Committee of the Primate Research Institute, Kyoto University (Permission Number: 2018-046), and all experiments were conducted according to the Guidelines for Care and Use of Nonhuman Primates established by the Primate Research Institute, Kyoto University (2010). For the second set of our experiments with Monkeys E and F on application of AAV2.1-A vector to chemogenetic manipulation, the experimental protocol was approved by the Animal Ethics Committee of the National Institutes for Quantum Science and Technology (Permission Number: 11-1038-11), and all experiments were conducted in accordance with the Guide for the Care and Use of Nonhuman Primates in Neuroscience Research (Japan Neuroscience Society; https://www.jnss.org/en/animal_primates). For the third set of our experiments with Monkeys G and H on application of AAV2.1-A vector to in vivo calcium imaging, the experimental protocol was approved by the Animal Experiment Committee of Osaka University (Permission Number: FBS-18-005), and all experiments were conducted according to the Guidelines for Animal Experiments established by Osaka University.

## Viral vector production

All AAV vectors were produced by the helper-free triple transfection procedure. Briefly, AAV-293 cells (70% confluent in Corning Cell Stack 10 chamber) were transfected by genome, helper (pHelper; Agilent Technologies, California, USA), and packaging plasmids (pAAV-RC1, pAAV-RC2, pAAV-RC5, and pAAV-RC9) with polyethylenimine (PEI Max; Polysciences, Warrington, USA). For production of AAV2.1-A or AAV2.1-B vector, the pAAV-RC1 plasmid-coding AAV1 capsid protein and the pAAV-RC2 plasmid-coding AAV2 capsid protein were transfected respectively with the ratio of 1:9 or 5:5[29–31]. The AAV1, AAV2.1-A/B, and AAV2 vectors were then purified by affinity chromatography (GE Healthcare, Chicago, USA), and concentrated to 150 µl by ultrafiltration (Amicon Ultra-4 10 K MWCO; Millipore, Billerica, USA). AAV5 and AAV9 vectors were purified by CsCl gradient and concentrated to the same volume by ultrafiltration. The titer of each vector stock was determined by quantitative PCR using Taq-Man technology (Life Technologies, Waltham, USA). Viral solutions were diluted in 0.1 M phosphate-buffered saline (PBS; pH 7.4) containing 0.001% Pluronic F-80 (Sigma-Aldrich, St. Louis, USA) prior to injection.

The transfer plasmid (pAAV-CMV-mKO1-WPRE) was constructed by inserting the mKO1 gene (MBL Lifescience, Tokyo, Japan) and the WPRE sequence into an AAV backbone plasmid (pAAV-CMV; Agilent Technologies). The pAAV-hSyn-mKO1-WPRE plasmid was constructed by replacing the CMV promoter with the human Synapsin promoter, and, further, the pAAV-Syn-hM3Dq-WPRE plasmid was constructed by replacing the mKO1 gene with the hM3Dq gene. The method for construction of the pAAV-Syn-hM3Dq-IRES-AcGFP-WPRE plasmid was reported elsewhere[13,38], and the pAAV-CMV-hM3Dq-IRES-AcGFP-WPRE plasmid was constructed by promoter replacement as described above. The pAAV-CaMKIIα/hSyn-GCaMP6s-WPRE plasmid was constructed by replacing the CMV promoter and the mKO1 gene of the pAAV-CMV-mKO1-WPRE plasmid with the mouse CaMKIIα promoter (0.4 k base pair) or human synapsin promoter and the GCaMP6s gene.

## Cortical injections of viral vectors

Four monkeys (Monkeys A-D in Table 1) were initially sedated with ketamine hydrochloride (5 mg/kg, i.m.) and xylazine hydrochloride (0.5 mg/kg, i.m.), and then anesthetized with sodium pentobarbital (20 mg/kg, i.v.) or propofol (5 mg/kg/hr). An antibiotic (Ceftazidime; 25 mg/kg, i.v.) was administered at the initial anesthesia. After the removal of a skull portion over the frontal lobe, twelve different types of AAV vectors expressing RFP were injected bilaterally into the medial wall of the frontal lobe by the aid of a magnetic resonance imaging (MRI)-guided navigation system (Brainsight Primate; Rogue Research, Montreal, Canada). The injections were made through a 10-µl Hamilton Neuros Syringe (Hamilton, Reno, USA) at the rate of 0.2 µl/min (0.5 µl/site, two sites/track). The vector injections for eight tracks were performed in the case of Monkey A and B, and those for twelve tracks were also performed in the case of Monkey C and D. The tracks were placed at least 4.5 mm apart from each other, and no overlapping of adjacent injections was detected on any occasion, as also evidenced by the distribution gaps of transgene-expressing neurons. After the experiments, the monkeys were monitored until full recovery from the anesthesia. The rats were anesthetized with 1.5–2.5% isoflurane and fixed in a stereotaxic frame (SR-10R-HT, Narishige, Tokyo, Japan). A volume of 0.5 µl of the AAV2.1-A, 1, 2, 5, and 9 vectors expressing RFP was injected unilaterally into the motor cortex in a single needle penetration for each animal. The anteroposterior, mediolateral, and dorsoventral coordinates (mm) relative to the bregma and dura were A: 1.0, L: 2.3, H: 1.0. The injections were performed at the rate of 0.1 µl/min through a 1-µl Hamilton Neuros Syringe (Hamilton) connected to a micropump (Legato100, KD Scientific, MA, USA).

All experimental procedures were performed in a special laboratory (biosafety level 2) designated for in vivo animal infectious experiments that had been installed at the Primate Research Institute, Kyoto University. Throughout the entire experiment, the animals were kept in individual cages that were placed inside a special safety cabinet in controlled temperature (23–26 °C) and light (12-hr on/off cycle) conditions. The animals were fed regularly with dietary pellets and had free access to water. Every effort was made to minimize animal suffering.

## Chemogenetic manipulation

Under general anesthesia as described above, five distinct types of AAV vectors carrying the hM3Dq gene were injected bilaterally into the striatum (i.e., the caudate nucleus and putamen) in two monkeys (Monkeys E and F in Table 1). Stereotaxic coordinates of the injected sites were defined based on overlaid magnetic resonance (MR) and computed tomography (CT) images created by PMOD image analysis software (PMOD Technologies, Zurich, Switzerland). The injections were made through a 10-µl Hamilton microsyringe in a single needle penetration for each vector (3.0 µl/site, one site/track, at the rate of 0.5 µl/min). The tracks were placed at least 3.0 mm apart from each other, and no overlapping of adjacent injections was detected on any occasion, as also evidenced by the distribution gaps of transgene-expressing neurons. PET imaging was performed over a period of several months to one year using the procedures described previously[38]. Briefly, the monkeys were sedated with ketamine hydrochloride (5 mg/kg, i.m.) and xylazine hydrochloride (0.5 mg/kg, i.m.), and the anesthetized condition was maintained with isoflurane (1–2%, inhalation) during the PET imaging. PET scans were done with micro-PET Focus220 scanner (Siemens Medical Solutions, Malvern, USA). Following transmission scans, emission scans were acquired for 90 min after intravenous bolus injection of [11C]DCZ (331.5-404.9 MBq) or [18F]FDG (233.3-306.9 MBq). Pretreatment with DCZ (1 µg/kg; Med-ChemExpress, Monmouth Junction, USA) or vehicle (1–2% dimethyl sulfoxide (DMSO) in 0.1-ml saline, without DCZ) was carried out 1 min before the [18F]FDG injection. The PET imaging data were reconstructed with filtered back-projection with attenuation correction. Voxel values were converted to standardized uptake values (SUVs) that were normalized by injected radioactivity and body weight using PMOD. Volumes of interest (VOIs) were manually drawn on the center of the injection site and the cerebellum using PMOD, by referring to MR images of individual monkeys. To estimate the specific binding of [11C]DCZ, the regional binding potential relative to non-displaceable radioligand ($BP_{ND}$) was calculated with an original multilinear reference tissue model using the cerebellum as a reference region[51]. For FDG-PET analysis, dynamic SUV images were motion-corrected and

then averaged between 60 and 90 min after the radioligand injection. The SUV ratio (SUVR) of voxel value was calculated as a percentage of the averaged value of the whole brain for comparison across the scans.

## In vivo calcium imaging

In the remaining two monkeys (Monkeys G and H in Table 1), we performed in vivo intrinsic signal optical imaging (ISOI), wide-field calcium imaging (WFCI), and two-photon calcium imaging (TPCI) of neurons in layers 2 and 3 of the V1 and V2. Under sterile conditions and isoflurane anesthesia (1–2%, inhalation), the cortical surface of V1 and V2 was exposed by incision of the scalp and removal of a skull portion and dura mater over these areas. The AAV2.1-A vector carrying the GCaMP6s gene was injected unilaterally into the V1 and V2 at six sites (1 μl at each site) in each monkey. For Monkey G, AAV2.1-A-CaMKIIα-GCaMP6s was injected into two loci at the titer of $2.0 \times 10^{13}$ gc/ml. The same vector was injected into the other two sites at the titer of $1.0 \times 10^{14}$ gc/ml. In addition, AAV2.1-A-Syn-GCaMP6s was injected into two loci at the titer of $2.0 \times 10^{13}$ or $1.0 \times 10^{14}$ gc/ml. For Monkey H, AAV2.1-A-CaMKIIα-GCaMP6s was injected into six loci at the titer of $4.0 \times 10^{13}$ gc/ml. The distance between the injection sites was at least 1.5 mm apart from each other. At each site, two injections were made at the depth of 1 and 2 mm (0.5 μl at each depth). After the vector injections, a chronic titanium imaging chamber (inner diameter, 12 mm) was implanted to cap the exposed cortex. The center of the chamber was positioned 15-mm lateral to the midline, and its anteroposterior position was adjusted so that the lunate sulcus ran along the one-third upper position of the chamber to allow a view of both V1 and V2[52]. The exposed cortex was covered with artificial dura (W. L. Gore & Associates, Inc., Newark, USA) that was glued to the chamber with silicone adhesive (Kwik-Sil; World Precision Instruments, Sarasota, USA) for preventing regenerative tissue from infiltrating into the imaging field[53]. The exposed cortex was further protected with a cover glass (Matsunami, Osaka, Japan; 14 mm in diameter and 0.5 mm in thickness), which was fixed to the chamber with a retaining ring (SM14RR; Thorlabs, Newton, USA). The retaining ring was glued to the titanium chamber with surgical silicone adhesive (Kwik-Sil).

When we performed imaging experiments, anesthesia was first introduced to the monkeys with ketamine and then maintained with propofol (induction, 5 mg/kg; maintenance, 5 mg/kg/hr, i.v.)[52]. The monkeys were paralyzed with vecuronium bromide (induction, 0.25 mg/kg, i.v.; maintenance, 0.05 mg/kg/hr, i.v.) and artificially respirated. The eyes were fitted with contact lenses of appropriate curvature to prevent from drying and focus on a stimulus screen 57-cm distant from the cornea[54,55]. In ISOI, intrinsic signals were acquired by a CMOS camera tandem-lens setup (FLIR GS3-U3-41C6NIR-C; exposure time, 100–300 μs; aperture, f5.6; lens combination, Noct-Nikkor 58 mm for the object side and Nikkor 50 mm for the CMOS side; Nikon, Tokyo, Japan) with 625-nm illumination (M625L3; Thorlabs) at a frame rate of 25 frames/s. One single imaging trial lasted 5 s (1 s before visual stimulus onset and 4 s after onset). The image size was $1024 \times 1024$ pixels representing a $12.5 \times 12.5$ mm² field of view, which covered the full field of the cortical surface within the imaging chamber. Images were acquired with custom-built software made with LabVIEW (National Instruments, Austin, USA). The setup for WFCI was similar to that for ISOI and shared the same imaging window and software. Calcium signals were acquired by the same CMOS camera (exposure time, 10-30 ms) through 500-nm-long pass filter (FELH0500; Thorlabs) under 490-nm illumination (M490L4; Thorlabs) at a frame rate of 25 frames/s. In TPCI, calcium signals were acquired with a two-photon excitation microscope (MOM with 8 kHz resonant scanner; Sutter, Novato, USA) equipped with a 16×, NA 0.8 water immersion objective lens (CFI175-LWD-16XW; Nikon) and 920-nm excitation laser (Insight X3; Spectra Physics, Milpitas, USA). The frame rate was 30.9 frames/s. The size of image areas was $512 \times 512$ pixels representing a $630 \times 630$ μm² field of view.

Visual stimuli were created using ViSaGe Mk II (Cambridge Research Systems, Rochester, UK) and displayed on a gamma-calibrated 23-inch LCD monitor (MDT231WG; Mitsubishi Electronic, Tokyo, Japan). We presented full-screen drifting black-white square-wave gratings. The duty cycle of gratings was 0.2 (20% white), and the spatial frequency was 1.5 cycle/degree. The gratings drifted at 8°/s and were presented in a randomly interleaved fashion in four orientations (0°, 45°, 90°, and 135°) with two opposite moving directions (i.e., eight directions; 0°, 45°, 90°, 135°, 180°, 225°, 270°, and 315°). Adding a blank screen condition, there were nine stimulus conditions in total. Each stimulus condition lasted 4 s. The initial phase of the gratings was randomly selected.

In data analysis, the magnitude of fluorescence response ($\Delta F/F$) was first quantified using the following formula: $\Delta F_i/F = (F_i - F_0)/F_0$, in which $F_i$ was a response in an $i$th single frame after stimulus onset, and $F_0$ was an averaged response of frames over a period of 1 s before stimulus onset. We used this fluorescence response to describe the response time course of a population of single neurons. In ISOI and WFCI, we compared responses to orthogonal orientations by performing subtraction of the averaged $\Delta F/F$ between two orthogonal orientation conditions. For this analysis, responses to opposite directions with the same orientation gratings were averaged. All image processing procedures were done with MATLAB (Mathworks). Blood vessel mask was created based on the cortical image under 530-nm illumination (M530L3; Thorlabs).

## Histology and image acquisition

The six monkeys (Monkeys A-F) and 20 rats underwent perfusion-fixation. The approximate survival periods after the vector injections were four weeks for Monkeys A-D, over six months for Monkey E, over 12 months for Monkey F, and three weeks for the rats. In Monkeys E and F, DCZ was administered systemically 2 hr prior to sacrifice for analysis of c-fos expression. The animals were anesthetized deeply with an overdose of sodium pentobarbital (50 mg/kg, i.v.) and perfused transcardially with 0.1 M PBS, followed by 10% formalin in 0.1 M phosphate buffer. The removed brains were postfixed in the same fresh fixative overnight at 4 °C and saturated with 30% sucrose in 0.1 M PBS at 4 °C. Coronal sections were cut serially at the 50-μm (monkeys) or 40-μm (rats) thickness on a freezing microtome and grouped into ten series. Every tenth section was used for individual histological analyses.

Double fluorescence histochemistry for NeuN and S100β was performed for Monkeys A-D and rats. Briefly, coronal sections were incubated with a cocktail of guinea pig polyclonal antibody against NeuN (1:1,000; Millipore) and mouse monoclonal antibody against S100β (1:5,000; Sigma-Aldrich). These sections were then incubated with a cocktail of DyLight 405-conjugated donkey anti-guinea pig IgG antibody and Alexa 647-conjugated donkey anti-mouse IgG antibody (1:200; Jackson ImmunoResearch, West Grove, USA). For Monkeys E and F, double immunofluorescence histochemistry for GFP and DARPP-32, PV, or ChAT was performed. Coronal sections were incubated with chicken monoclonal antibody against GFP (1:2,500; Abcam, Cambridge, UK) and rabbit monoclonal antibody against DARPP-32 (1:250; Cell Signaling Technology, Danvers, USA), mouse monoclonal antibody against PV (1:2,000; Sigma-Aldrich), or goat monoclonal antibody against ChAT (1:200; Sigma-Aldrich). These sections were then incubated with Alexa 488-conjugated donkey anti-chicken IgG antibody (1:400; Jackson ImmunoResearch) and Alexa 555-conjugated donkey anti-rabbit antibody (1:400; Invitrogen, Waltham, USA), Alexa 555-conjugated donkey anti-mouse IgG antibody (1:400; Invitrogen), or Alexa 647-conjugated donkey anti-goat IgG antibody (1:400; Jackson ImmunoResearch).

For immunohistochemical staining for GFP, hM3Dq, and c-fos following chemogenetic manipulation, coronal sections were pre-treated with 0.3% $H_2O_2$ for 30 min, rinsed three times in 0.1 M PBS, and immersed in 1% skim milk for 1 hr. The sections were then incubated for

two days at 4 °C with rabbit monoclonal anti-GFP antibody (1:2,000; Invitrogen), rabbit polyclonal anti-M3 (muscarinic acetylcholine receptor M3) antibody (1:200; Atlas Antibodies, Stockholm, Sweden), or rabbit polyclonal anti-c-fos antibody (1:2,000; Abcam) in 0.1 M PBS containing 2% normal donkey serum and 0.1% Triton X-100. Subsequently, the sections were incubated with biotinylated donkey anti-rabbit IgG antibody (1:1,000; Jackson ImmunoResearch) in the same fresh medium for 2 hr at room temperature, followed by the avidin-biotin-peroxidase complex (ABC *Elite*; 1:200; Vector laboratories, Burlingame, USA) in 0.1 M PBS for 2 hr at room temperature. The sections were finally reacted for 10-20 min in 0.05 M Tris-HCl buffer (pH 7.6) containing 0.04% diaminobenzine tetrahydrochloride (Wako, Osaka, Japan), 0.04% $NiCl_2$, and 0.002% $H_2O_2$. The reaction time was set to make the density of background immunostaining almost identical across different sections. All histological sections were mounted onto gelatin-coated glass slides and counterstained with 0.5% Neutral red.

A high-sensitivity charge-coupled device (CCD) camera (VB-7010, Keyence, Osaka, Japan) was used to capture bright-field and fluorescence images from the whole brain sample. To capture the bright-field and fluorescence images in coronal sections, a scientific CMOS (sCMOS) camera (In Cell Analyzer 2200; GE Healthcare, Chicago, USA) was used. We created merged images of mKO1 and NeuN/GFAP-immunofluorescent coronal sections by using Fiji/ImageJ software[56]. A confocal laser-scanning microscope (LSM800; Carl Zeiss, Oberkochen, Germany) was also used to take fluorescence images. We prepared merged images using the Zeiss Zen 3.3 software (Carl Zeiss).

### Intensity analysis of mKO1 native-fluorescence at injection sites

The intensity of mKO1 native-fluorescence emitted from each vector injection site was measured in sections of the monkey brains by using MATLAB software (Mathworks, Natick, USA). The region for measurement in each section was determined according to the highest one selected from the averaged intensity values within circular observation windows (2-mm diameter for monkeys and 1-mm diameter for rats). Three sections were chosen including the sections through the center of the injection site (revealed by the highest fluorescence intensity) and 500 μm anterior/posterior to the injection site. In each case, intensity measurement at the injection site was performed by setting circles with the highest intensity, and the averaged value of fluorescence intensity obtained in the three sections was calculated to determine the relative transgene expression level for each vector. To estimate the transgene expression level at each injection site, the expression level for the AAV2-CMV or AAV5-CMV vector was defined as 1.0, and the relative value to this was calculated for the other vectors. The same procedure was done for the intensity analysis in the rat brain, except that three sections measured were 400 μm apart.

### Neuron specificity analysis of transgene-expressing cells at injection sites

The number of RFP-positive cells expressing NeuN or S100β was counted at the injection sites of the vectors in Monkeys A-D and the rats. Three sections were chosen in the same fashion as described above. Stereological cell counts assisted with Stereo Investigator software (MBF Biosciences, Williston, USA) were carried out in $100 \times 100 \mu m^2$ counting frames equally spaced across a grid of $250 \times 250 \mu m^2$ (monkeys) or $150 \times 150 \mu m^2$ (rats) with 18-μm-high optical dissector (average section thickness, 25 μm for monkeys, 20 μm for rats). The cell counts were done over the whole transgene-expressing region in each section. Based on these data, the ratio of double-labeled cells to the total RFP-positive cells was calculated.

### Counts of mKO1- and c-fos-positive neurons

For counts of mKO1-positive neurons at the cortical injection sites, three sections were selected in the same manner as described above.

The number of mKO1-positive neurons was counted by the aid of Stereo Investigator software (MBF Biosciences). For counting the number of thalamic neurons via retrograde or anterograde trans-neuronal labeling, three equidistant (500 μm apart) sections were selected in rats, and similar stereological cell counts were carried out in $250 \times 250 \mu m^2$ counting frames equally spaced across a grid of $750 \times 750 \mu m^2$ with an 18-μm-high optical dissector (average section thickness, 20 μm). For counts of c-fos-positive neurons in the striatum, Neurolucida software (MBF Biosciences) was used to set circular observation windows with the densest labeling (1-mm diameter), and the averaged value of cell counts was calculated.

### Statistics

Expressed as the mean ± SEM of the data were the normalized fluorescence intensity and the number of RFP-positive neurons for all quantifications in the rat. For statistical comparison, one-way analysis of variance followed by post hoc Dunnett's test was used with significance set at $*P < 0.05$, $**P < 0.01$, $***P < 0.001$.

### Reporting summary

Further information on research design is available in the Nature Portfolio Reporting Summary linked to this article.

## Data availability

The regents and antibodies used in this study are listed in Supplementary Data 1. The viral vectors used in this study are available from the corresponding authors upon reasonable request. Source data are provided with this paper.

## Code availability

The code to generate the results of this study is available at Zenodo: (https://doi.org/10.5281/zenodo.8137537).

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

## Acknowledgements

We are grateful to K. Nagaya, E. Tanaka, E. Sumiya, S. Nonomura, T. Okauchi, R. Suma, Y. Sugii, R. Yamaguchi, Y. Matsuda, J. Kamei, and N. Nitta for their technical assistance. We also thank Dr. M.-R. Zhang and his colleagues at the Department of Advanced Nuclear Medicine Sciences, QST, for producing the radioligand. This work was supported by MEXT/JSPS KAKENHI Grant Numbers JP19F193830 to Y.F., JP14J01649 to R.T., JP18H05007, and JP21H02596 to I.F., JP19H03335 and 22H05157 to K.I., and JP19H05467 to M.T., by AMED Grant Numbers JP18dm0307007 and JP16dm0107146 to T.M., JP20dm0307021 to K.I., and JP21dm0207077 to M.T., and by JST Grant Numbers JPMJCR1683 to K.I. and JPMJMS2295 to T.M. and K.I.

## Author contributions

K.I. and M.T. designed the experiments. M.F., M.N., and K.I. prepared the viral vectors. K.K., S.T., A.Z., and K.I. performed experiments on transgene expression pattern analysis of mosaic vectors. K.K., Y.N., Y.H., and T.M. performed experiments on the application of mosaic vectors to chemogenetic manipulation. G.H., Y.F., R.F.T., M.I., I.F., and K.I. performed experiments on the application of the mosaic vectors to in vivo calcium imaging. K.K., Y.N., and G.H. analyzed the data. K.K., T.M., I.F., K.I., and M.T. wrote the manuscript. All authors discussed the data and commented on the manuscript.

## Competing interests

The authors declare no competing interests.
