## [Peer Review File · Nature Communications]

A mosaic adeno-associated virus vector as a versatile tool that exhibits high levels of transgene expression and neuron specificity in primate brainReviewers' comments:

Reviewer #1 (Remarks to the Author):

This set of studies aims to solve some of the problems facing gene transduction in nonhuman primates. They have created a novel AAV vector, AAV2.1, which appears to show the sensitivity of the AAV1 vector along with the specificity of the AAV2 vector. This potentially combines the best features of both. They go one step further and demonstrate the utility of the virus for chemogenetics and calcium imaging.

This is a relatively straightforward paper. I have only minor quibbles with the results and interpretation. I think the significance may be somewhat limited, however, because this construct was not compared to the commonly used AAV5 and AAV9, and because the community of NHP virus users is relatively limited. Maybe this vector would be promising for human gene therapies, which could enhance the significance? The significance is my only major concern.

Minor comments

Separating Figures 3 and 5 because they are in different subjects made comprehension more difficult for me as a reader. I would suggest combining these two figures and weaving their textual explanations together, with an emphasis on what was the same vs different across subjects.

Has Monkey D not been sacrificed yet, and that's why there is no histology to match Figure 4? It's fine, but just needs to be explained.

In Figure 4A, it looks like there's some surface+needle track leakage for the AAV2.1 injections (on the left), but not for the other 2 injections. Can this be discussed? Might the sensitivity of this virus mean it carries a higher probability of contamination?

I appreciate the prolonged timecourse of these experiments, it's something that's of concern but there hasn't been a great deal of data out there.

Reviewer #2 (Remarks to the Author):

This manuscript describes a hybrid AAV serotype obtained by mixing capsid proteins from AAV1 and AAV2, developed with the aim of increasing the neuronal specificity and penetrance in non-human primate (NHP) experiments. Two versions of this mosaic AAV (named AAV2.1-A (AAV1 =10%/ AAV2 = 90%), and AAV2.1-B(AAV1=50%/AAV2=50%)) were initially tested in 2 macaques. AAV2.1-A showed higher neuronal specificity, particularly when the transgene expression was driven by the neuronal promoter Synapsin. AAV2.1-A was then used to transduce the excitatory DREADD hM3Dq and the calcium sensor CaMKIIa in monkeys. The results demonstrate robust, long lasting, and functional expression of the transgene.

The study has many strengths: The effectiveness of AAV2.1-A was tested to deliver various transgenes (mKO1, hM3Dq and CaMKIIa), and the transgene expression was evaluated along the course of many months. The expression patterns of the mosaic were compared side by side with more traditional AAVs, and various of promoters and tag proteins were tested.

Given the current obstacles to further expand the use of genetic-based approaches in NHP research, the results of this study are exciting and promising, since AAV2.1-A could provide a useful tool to advance these approaches in NHP research. This could be a significant advance to the field. However, there are some important gaps in the experimental design, analysis and data interpretation of the study, that should be addressed further strengthen the significance and potential of this novel virus.

- A major concern is the relatively large number of different viral vectors tested per monkey, and the consequent problem of reliably identifying the expression produced by each vector. Monkeys A and B received 8 different viruses, all carrying the fluorescent protein mKO1 as transgene. It may be difficult to clearly identify the pattern and extent of transgene expression for each virus. Can it be assessed, with certainty, what are the limits of the volume of tissue transduced by each virus? Do the expression fields of each virus overlap? For example, in figure 1a (lower panel), the fluorescent-positive regions originated by injections 3 and 4 seem to overlap.
- Furthermore, multiple viral injections make it difficult to assess which of the virus may have originated anterograde or retrograde labeling. For example, if there are mKO1-positive axonal terminals in subcortical structures of monkeys A and B, how could it be determined which of the cortical injections gave rise to these terminals?
- The same concern exists for experiments in monkeys C and D. How was the area of each individual injection defined? Was there overlap between injections? In this case, the injections were 3 ul each in different compartments of the striatum, thus presumably the internal capsule could provide a natural barrier to prevent the spread of the solutions between caudate and putamen, but it should be clearly indicated how the extent of each injection was defined.
- A detailed analysis of anterograde or retrograde transport was not conducted for any of the animals that were euthanized. Retrograde transport of the viral particles and even transsynaptic transport are possible, and expression of the transgene at axon terminals should be examined. A thorough characterization of transgene expression in cell bodies and axonal terminals throughout the brain (or at least in areas known to be connected to the regions that received virus in monkeys) A, B, C and D is needed.
- The analysis to quantify the intensity of transgene expression in animals A and B is critical for the main conclusion of this study, and for the experiments done in the rest of the animals. The current analysis, based on intensity of the mKO1 is unsatisfactory. The analysis should include a clear description of how the center of the injection site was identified, the actual number of mKO1-positive neurons and the total number of neurons counted, not only the proportions as currently shown in fig 2b.
- A marker other than GFAP should be used to identify astrocytes, since some astrocytes do not express GFAP, or express it at low levels (see, for example, Yu, X., et al. (2020). "Improved tools to study astrocytes." *Nat Rev Neurosci* 21(3): 121-138.).
- The histological analysis for animals in the chemogenetic study (C and D) should be more thorough, including:
 - o A descriptions of anterograde and retrograde transport. The current version of the manuscript mentions lack of GFP positive neurons in the substantia nigra compacta, but the SNc is only one of many targets that project to the striatum. In fact, figure 3d shows FDG uptake signal in motor cortices, which would indicate retrograde transport to cortices from the striatum.
 - o Quantification of transduced neurons (similar to what was done for monkey A and B). The current version of the manuscript provides only a qualitative evaluation of the efficacy of the vectors (see lines 195-197 ("much denser or broader"))
 - o Assessment of specificity (neurons or glia), and a description of the types of neurons transduced. Furthermore, some effort should be done to identify the types of striatal neurons transduced. Relevant to this analysis is that the type of neurons (projections or interneurons) transduced could be influenced by the interactions between the capsid used and the promoter (Powell et al 2020; Bohlen et al 2020).
 - o Provide higher magnification examples of GFP+ cells in animal C (fig 4). There is a robust GFP-positive neuropil staining shown in fig. 4b. What could be the sources of this neuropil?
- Histology data for monkey D is missing. It should be included, containing similar analysis as done for monkey C.
- In regards to the calcium imaging experiment, the text states that "...the first monkey (Monkey E in Table 1) was used primarily for determining optimal experimental conditions" (lines 237-238) No data is shown for monkey E, although the Methods section indicates that monkey E was used for imaging. Thus, data from monkey E should be shown too.

Some additional minor points to consider:

- The introduction should mention previous work where mosaic AAVs have been used to improve or modify the efficiency of AAV serotypes
- For all animals, indicate what is the distance between injections
- Indicate in which cortical regions were the injections in animals A and B performed.
- In the cases where the same virus was used, but at a lower titer, indicate how were virus solutions diluted
- Line 180: define 'sufficiently high'
- Figures 3 and 5 should be consolidated. It would make it easier to compare the results in the 2 animals.
- In panel A of fig 3., please indicate in the drawing that the virus contained GFP. This would make it more comparable to the panel 5A
- In Discussion, lines 288-289 state "...contained in this vector, because the production efficiency of AAV1 vector is relatively high to that of AAV2 vector". This statement is unclear. Does this refer to the production of the transgene? Why does the AAV1 vector have higher production efficiency than AAV2? Please clarify
- One of the many interesting findings of this study is that the expression of DREADDs, at least based on the extent of FGD uptake after injections with AAV2 vector without GFP is comparable to that obtained using AAV2.1-A with GFP. This observation underscores that the fluorescent proteins could limit the expression of the receptors of interest. The discussion should mention that other non-fluorescent tag proteins are available.
- In Methods, please add the following information: ages of animals at the time the study started, distance between sites of injections for all cases, injection rates for all cases, describe how was 'the brightest circle' (line 541) identified.
- In figure 2, please indicate if each bar represents the average of the 3 sections used for the quantification
- In legend to figure 5b, clarify if the calcium transient are spontaneous or in response to visual stimuli.

Reviewer #3 (Remarks to the Author):

In this paper, Kimura and colleagues develop a novel AAV serotype to enhance expression levels in the primate brain while limiting expression in glia cells. While such a vector would be of good use for the primate neuroscience community, the paper in its current form is in my opinion not suitable for publication in Nature Communications. The major reason for this recommendation is the lack of an adequate number of repeats leading to a data set that is simply too limited to support the conclusions. While it is unfortunately quite common that virus tests in primates rely on very few injections per virus, spread out over different brain areas, it is exactly this lack of repeated testing that is making it complicated for labs to identify the most suitable virus. There is sufficient variation between virus injections of the same construct, into the same brain region, that 2 injections per virus are insufficient to determine expression levels reliably. The authors' own data (e.g. Figure 1C, Figure 2B) clearly illustrate this point in terms of expression levels, and even expression patterns (neurons versus glia activation). In addition, it is also often observed that the same virus, injected into different brain areas, can result in quite different expression patterns e.g. because of differences in tropism. For this study, this means that not only is the sample size insufficient to assess each virus individually, the comparison between serotypes is fundamentally flawed because they were never injected into the same part of the brain. Lastly, while it is useful to quantify expression patterns by the number of neurons or glia cells that were infected, the expression intensity seems like a poor choice (because it, as the authors acknowledge, be influenced by other factors like which cells are expressing the strongest). It would be much more meaningful to quantify the radius of expression, laminar distribution, and neuron density per injection.

We are grateful to the three reviewers for their critical reading of our manuscript and providing us with invaluable and insightful comments. Here are point-by-point responses to the reviewers' comments.

Reviewer 1

Major comment

1. I have only minor quibbles with the results and interpretation. I think the significance may be somewhat limited, however, because this construct was not compared to the commonly used AAV5 and AAV9, and because the community of NHP virus users is relatively limited. Maybe this vector would be promising for human gene therapies, which could enhance the significance? The significance is my only major concern.

We well understand the reviewer's criticism. As the reviewers pointed out, the significance of the mosaic AAV2.1-A vector was limited in our original manuscript, because (1) we did not compare its gene transduction properties with those of AAV5 and AAV9, both which have widely been used in neuroscience research and (2) we characterized the AAV2.1-A vector only in the primate brain, but not in the rodent brain that has been a research subject for many more neuroscientists. In line with the reviewer's comment, we have performed additional series of experiments with two monkeys and 20 rats, and provided data about the comparison of gene transduction properties (transgene expression intensity and neuron specificity) among the AAV2.1-A, AAV1, AAV2, AAV5, and AAV9 vectors (see Figs. 3,4,5). These additional experiments and data analyses augment the superiority of our mosaic vector, AAV2.1-A vector, to the other AAV vectors and underscore the utility and versatility of this vector across the animal species (for Results, see page 7, line 2 from the bottom through page 9, line 4 from the bottom; for Discussion, see page 16, lines 5-17 from the bottom). We now believe that such additional data provided in our revised manuscript greatly contribute to enhance the significance of the mosaic vector and make it more beneficial to broader readers.

Minor comments

1. Separating Figures 3 and 5 because they are in different subjects made comprehension more difficult for me as a reader. I would suggest combining these two

figures and weaving their textual explanations together, with an emphasis on what was the same vs different across subjects.

According to the reviewer's suggestion, Figures 3 and 5 of the original manuscript have been combined, as Figure 6 of the revised version, for readers' easier comprehension and partly modified the textual explanations.

2. Has Monkey D not been sacrificed yet, and that's why there is no histology to match Figure 4? It's fine, but just needs to be explained.

In line with the reviewer's comment, we have added histological data, not shown in the original manuscript, obtained in Monkey D (Monkey F of the revised version) and combined them with the data in Monkey C (Monkey E of the revised version) as new Figure 7 (see also page 12, line 4 from the bottom through page 13, line 10 from the bottom).

3. In Figure 4A, it looks like there's some surface+needle track leakage for the AAV2.1 injections (on the left), but not for the other 2 injections. Can this be discussed? Might the sensitivity of this virus mean it carries a higher probability of contamination?

As pointed out by the reviewer, it is indeed needle track leakage in the AAV2.1 vector injection. However, such leakage happened not only for the AAV2.1 vector, but also for the other vectors (as seen in different rostrocaudal levels of cross sections), though the intensity of cortical leakage of each vector seems parallel to that of the striatal injection (data not shown here). This implies that the leakage does not necessarily reflect properties of the vectors. While the histological data obtained in Monkeys E and F have been combined in the revised version as mentioned above, panels showing the vector injection sites in the striatum have been re-arranged for readers' easier comparison (see Fig. 7a,b).

Reviewer 2

Major comments

1. A major concern is the relatively large number of different viral vectors tested per monkey, and the consequent problem of reliably identifying the expression produced by each vector. Monkeys A and B received 8 different viruses, all carrying the fluorescent protein mKO1 as transgene. It may be very difficult to clearly identify the pattern and

extent of transgene expression for each virus. Can it be assessed, with certainty, what are the limits of the volume of tissue transduced by each virus? Do the expression fields of each virus overlap? For example, in figure 1a (lower panel), the fluorescent-positive regions originated by injections 3 and 4 seem to overlap.

We well understand the reviewer's concern. Although some of the vector injections into the cerebral cortex seemed to overlap because of fluorescent halation and anterograde axonal labeling, we confirmed that the extent of injection sites of the AAV vectors was a maximum of 1.5 mm in radius, and, in fact, there were distinct gaps between neighboring injections. The existence of such gaps was confirmed with the distribution of transgene-expressing neurons in our cell count analysis. As we have added some descriptions in the Results and Methods sections (see page 6, lines 8-10; page 21, lines 6-8 from the bottom), the tracks were placed at least 4.5 mm apart from each other, and no overlapping of adjacent injections was detected on any occasion, as also evidenced by the distribution gaps of transgene-expressing neurons. Moreover, we have replaced fluorescence images showing the cortical injection sites of the vectors with line drawings to avoid misunderstanding due to fluorescent halation and anterograde axonal labeling (see Figs. 1a and 3a).

2. Furthermore, multiple viral injections make it difficult to assess which of the virus may have originated anterograde or retrograde labeling. For example, if there are mKO1-positive axonal terminals in subcortical structures of monkeys A and B, how could it be determined which of the cortical injections gave rise to these terminals?

We well understand the reviewer's point. Since only the intensity analysis of transgene expression could not discriminate anterograde axonal labeling through corticocortical connections in each injection site, we have further performed counts of transgene-expressing neurons to avoid any bias for transgene expression analysis in Monkeys A-D. Consequently, strikingly similar tendencies of transgene expression were obtained between the intensity analysis of the vectors with Syn promoter and the counting analysis of transgene-expressing neurons (see Figs. 1c,d and 3c,d; page 7, lines 2-6; page 8, lines 8-11 from the bottom). We also confirmed that retrograde labeling with individual AAV vectors was only very rarely observed in Monkeys E and F in which the vectors were injected into the striatum (see Fig. 7g; page 13, lines 8-13). Moreover, in an additional series of experiments with 20 rats, only a few, if any, retrogradely labeled neurons were found in the cortex contralateral to the injection sites of individual AAV

vectors (data will be shown if necessary).

3. The same concern exists for experiments in monkeys C and D. How was the area of each individual injection defined? Was there overlap between injections? In this case, the injections were 3 ul each in different compartments of the striatum, thus presumably the internal capsule could provide a natural barrier to prevent the spread of the solutions between caudate and putamen, but it should be clearly indicated how the extent of each injection was defined.

We well understand the reviewer's concern. Since the striatal injection of each vector was carried out only in a single needle penetration in these monkeys (Monkeys E and F of the revised version), the injection site was so limited as to ensure no possible overlapping between the injections into the caudate nucleus and the putamen on the same side. In fact, the distance between the injection sites in the caudate nucleus and the putamen was far enough (at least 3 mm on the mediolateral plane) to prevent their mutual contamination (see page 22, lines 3-5 from the bottom). As the reviewer presumed, the internal capsule appears to provide a natural barrier to prevent the spread of the injections (see Fig. 7a,b).

4. A detailed analysis of anterograde or retrograde transport was not conducted for any of the animals that were euthanized. Retrograde transport of the viral particles and even transsynaptic transport are possible, and expression of the transgene at axon terminals should be examined. A thorough characterization of transgene expression in cell bodies and axonal terminals throughout the brain (or at least in areas known to be connected to the regions that received virus in monkeys) A, B, C and D is needed.

We well understand the reviewer's point. Because of the limitation of the number of monkeys used for the present study, multiple AAV vectors were injected bilaterally into frontal cortical areas of single monkeys (Monkeys A-D). Therefore, it was unfortunately impossible to distinguish anterograde/retrograde labeling among the vectors in these monkeys. Instead, in Monkeys E and F in which the vectors were injected into the striatum (especially in the case of AAV2.1-A vector injections into the caudate nucleus and putamen on the same side in Monkey E; see Fig. 7g; page 13, lines 8-13; see also the legend of Fig. 7g), we indeed confirmed that robust anterograde axonal labeling was distributed in the output regions of the striatum (i.e., the substantia nigra pars reticulata and the external and internal segments of the globus pallidus). By contrast, only very

rarely was retrograde labeling seen in the input regions of the striatum (i.e., the substantia nigra pars compacta and the thalamic motor nuclei). Robust anterograde but sparse retrograde labeling of the vectors was also verified in the rat brain. Since we fully understand the importance of histological analysis of axonal transport capacity of the AAV2.1-A vector in comparison with the other AAV vectors as the reviewer indicated, we would like to summarize the data, as the next work, obtained from the striatal injections of these vectors.

5. The analysis to quantify the intensity of transgene expression in animals A and B is critical for the main conclusion of this study, and for the experiments done in the rest of the animals. The current analysis, based on intensity of the mKO1 is unsatisfactory. The analysis should include a clear description of how the center of the injection site was identified, the actual number of mKO1-positive neurons and the total number of neurons counted, not only the proportions as currently shown in fig 2b.

We totally agree with the reviewer's criticism. As described in the Response to Major Comment 2, we have added cell-count analysis at the center of each vector injection site, revealed by the highest fluorescence intensity (page 29, line 1-3), in Monkeys A-D. Data obtained in these monkeys, including the actual number of transgene-expressing neurons, have been provided in Figures 1c,d and 3c,d (see also page 7, lines 2-6; page 8, lines 8-11 from the bottom).

6. A marker other than GFAP should be used to identify astrocytes, since some astrocytes do not express GFAP, or express it at low levels (see, for example, Yu, X., et al. (2020). "Improved tools to study astrocytes." Nat Rev Neurosci 21(3): 121-138.).

In line with the reviewer's comment, we have replaced data obtained from GFAP immunostaining with those from S100 β immunostaining for analyzing the neuronal vs. glial tropism in Monkeys A-D (see Figs. 2 and 4). Although we further tried to immunostain with antibodies against other glial markers, SOX9 and ALDH1L1, neither of them worked successfully in the monkey brain.

7. The histological analysis for animals in the chemogenetic study (C and D) should be more thorough, including:

• A descriptions of anterograde and retrograde transport. The current version of the manuscript mentions lack of GFP positive neurons in the substantia nigra compacta, but

the SNc is only one of many targets that project to the striatum. In fact, figure 3d shows FDG uptake signal in motor cortices, which would indicate retrograde transport to cortices from the striatum.

According to the reviewer's comment, we have added panels showing no or rare retrograde labeling in the thalamus as well as in the nigra (see Fig. 7g; page 13, lines 8-13; see also the legend of Fig. 7g). Although the FDG uptake signal emerged in the cortex, this was most likely due to vector leakage along the injection needle track (see also the Response to Minor Comment 3 raised by Reviewer 1). In fact, we failed to find labeled neurons frequently in cortical areas other than vector leakage sites.

• Quantification of transduced neurons (similar to what was done for monkey A and B). The current version of the manuscript provides only a qualitative evaluation of the efficacy of the vectors (see lines 195-197 ("much denser or broader").

We well understand the reviewer's concern. No quantitative evaluation of transgene expression levels of the vectors was carried out for the following three reasons: (1) Expression levels of hM3Dq, instead of fluorescent tag protein, were analyzed by DCZ-PET scans; (2) Fluorescence intensity of IRES-GFP was not strong enough to represent differential levels of transgene expression precisely; and (3) Immunostaining for hM3Dq (Monkey F) resulted in no labeling of neuronal cell bodies.

• Assessment of specificity (neurons or glia), and a description of the types of neurons transduced. Furthermore, some effort should be done to identify the types of striatal neurons transduced. Relevant to this analysis is that the type of neurons (projections or interneurons) transduced could be influenced by the interactions between the capsid used and the promoter (Powell et al 2020; Bohlen et al 2020).

In line with the reviewer's comment, we have added some data about identification of the types of striatal neurons transduced. As described in the Results section (see page 13, lines 2-9 from the bottom; Supplementary Fig. 1), we characterized striatal neurons in which the transgene was expressed via the AAV2.1-A vector by identifying their neuron types (i.e., projection neurons vs. interneurons). Double fluorescence histochemistry was performed to analyze the colocalization of dopamine- and cAMP-regulated phosphoprotein (DARPP-32) as a medium spiny projection neuron marker, or parvalbumin (PV)/choline acetyltransferase (ChAT) as fast-spiking/cholinergic

interneuron markers in GFP-positive cells. We found that the AAV2.1-A vector resulted in gene transduction into both projection neurons and interneurons in the striatum.

• Provide higher magnification examples of GFP+ cells in animal C (fig 4). There is a robust GFP-positive neuropil staining shown in fig. 4b. What could be the sources of this neuropil?

In response to the reviewer's comment, we have provided a higher-magnification photo of GFP-positive neurons, as well as anterogradely labeled fibers and terminals (which is the source of neuropil staining), in the nigra (see Fig. 7g).

8. Histology data for monkey D is missing. It should be included, containing similar analysis as done for monkey C.

According to the reviewer's comment, we have added histological data obtained in Monkey D (Monkey F of the revised version) and combined them with the data in Monkey C (Monkey E of the revised version) as new Figure 7 (see also page 12, line 4 from the bottom through page 13, line 10 from the bottom; see also the Response to Minor Comment 2 raised by Reviewer 1).

9. In regard to the calcium imaging experiment, the text states that "...the first monkey (Monkey E in Table 1) was used primarily for determining optimal experimental conditions" (lines 237-238) No data is shown for monkey E, although the Methods section indicates that monkey E was used for imaging. Thus, data from monkey E should be shown too.

According to the reviewer's comment, we have added calcium imaging data obtained in Monkey E (Monkey H of the revised version) as new Supplementary Figure 2.

Minor comments

1. The introduction should mention previous work where mosaic AAVs have been used to improve or modify the efficiency of AAV serotypes.

In line with the reviewer's suggestion, we have added some descriptions of a previous work on mosaic AAV vectors in the Discussion, but not the Introduction section (see page 15, lines 7-14 from the bottom).

2. For all animals, indicate what is the distance between injections.

In response to the reviewer's comment, we have described the distance between the cortical injections in the Results and Methods sections (see page 6, lines 8-10; page 21, lines 6-8 from the bottom; see also the Response to Major Comment 1). As described above, the distance between the injection sites in the caudate nucleus and the putamen was at least 3 mm on the rostrocaudal plane (see page 22, lines 3-5 from the bottom; see also the Response to Major Comment 3). The distance between the injection sites in the visual cortex was at least 1.5 mm apart from each other (see page 24, lines 4-5).

3. Indicate in which cortical regions were the injections in animals A and B performed.

In response to the reviewer's comment, we have described the actual cortical areas where vector injections were made in the Results section (see page 6, line 2).

4. In the cases where the same virus was used, but at a lower titer, indicate how were virus solutions diluted.

In response to the reviewer's comment, we have described how to dilute a viral solution in the Methods section (see page 20, lines 9-10 from the bottom).

5. Line 180: define 'sufficiently high'.

In response to the reviewer's comment, we have deleted this phrase not only here, but also in other corresponding parts.

6. Figures 3 and 5 should be consolidated. It would make it easier to compare the results in the 2 animals.

According to the reviewer's suggestion, Figures 3 and 5 of the original manuscript have been combined, as Figure 6 of the revised version, for readers' easier comparison (see also the Response to Minor Comment 1 raised by Reviewer 1).

7. In panel A of fig 3., please indicate in the drawing that the virus contained GFP. This would make it more comparable to the panel 5A.

In response to the reviewer's comment, we have modified Figure 6a in the revised version.

8. In Discussion, lines 288-289 state "...contained in this vector, because the production efficiency of AAV1 vector is relatively high to that of AAV2 vector". This statement is unclear. Does this refer to the production of the transgene? Why does the AAV1 vector have higher production efficiency than AAV2? Please clarify.

Since the corresponding part was judged to be described inappropriately, we have removed it from the Discussion section.

9. One of the many interesting findings of this study is that the expression of DREADDs, at least based on the extent of FGD uptake after injections with AAV2 vector without GFP is comparable to that obtained using AAV2.1-A with GFP. This observation underscores that the fluorescent proteins could limit the expression of the receptors of interest. The discussion should mention that other non-fluorescent tag proteins are available.

According to the reviewer's suggestion, we have added some descriptions of other non-fluorescent tag proteins in the Discussion section (see page 17, lines 9-12 from the bottom).

10. In Methods, please add the following information: ages of animals at the time the study started, distance between sites of injections for all cases, injection rates for all cases, describe how was 'the brightest circle' (line 541) identified.

In response to the reviewer's comments, we have added descriptions of ages of animals at the time the study started (see page 19, lines 4-7), distance between sites of injections for all cases (see page 21, lines 6-8 from the bottom; see also Figs. 1a and 3a), injection rates for all cases (see page 21, lines 11 from the bottom; page 22, lines 1-2) in the Methods section. Also, we have described how 'the brightest circle' was identified in the Methods section (see page 29, line 1-3).

11. In figure 2, please indicate if each bar represents the average of the 3 sections used for the quantification.

In response to the reviewer's comment, we have added the corresponding explanation to the legends of Figures 2 and 4.

12. In legend to figure 5b, clarify if the calcium transient is spontaneous or in response to visual stimuli.

In response to the reviewer's comment, we have added the corresponding explanation to the legends of Figure 8 and Supplementary Figure 2.

Reviewer 3

Major comments

In this paper, Kimura and colleagues develop a novel AAV serotype to enhance expression levels in the primate brain while limiting expression in glia cells. While such a vector would be of good use for the primate neuroscience community, the paper in its current form is in my opinion not suitable for publication in Nature Communications.

1. The major reason for this recommendation is the lack of an adequate number of repeats leading to a data set that is simply too limited to support the conclusions. While it is unfortunately quite common that virus tests in primates rely on very few injections per virus, spread out over different brain areas, it is exactly this lack of repeated testing that is making it complicated for labs to identify the most suitable virus. There is sufficient variation between virus injections of the same construct, into the same brain region, that 2 injections per virus are insufficient to determine expression levels reliably. The authors' own data (e.g. Figure 1C, Figure 2B) clearly illustrate this point in terms of expression levels, and even expression patterns (neurons versus glia activation). In addition, it is also often observed that the same virus, injected into different brain areas, can result in quite different expression patterns e.g. because of differences in tropism. For this study, this means that not only is the sample size insufficient to assess each virus individually, the comparison between serotypes is fundamentally flawed because they were never injected into the same part of the brain.

We seriously accept the reviewer's criticism and well understand a crucial pitfall of our present work on nonhuman primates. In line with the reviewer's point, we have performed additional series of experiments with two monkeys and 20 rats, and provided more data about gene transduction properties of the AAV2.1-A vector, which included the comparison of those of the AAV5 and AAV9 vectors, both of which have often been

utilized in prior studies (see Figs. 3 and 4; see also the Response to Major Comment 1 raised by Reviewer 1).

Moreover, we have depicted line drawings of the sites of vector injections in all monkeys (Monkeys A-D), instead of fluorescence images used in our original manuscript, to demonstrate repetitive injections of the AAV2.1-A vector into the monkey frontal cortex for analyzing appropriately its gene transduction properties, i.e., transgene expression and neuron specificity (see Figs. 1a and 3a). As shown in these drawings, individual vectors inserted with different promoters were injected almost symmetrically into cortical areas of both hemispheres in Monkeys A and B. In Monkeys C and D, the sites of cortical injections of the AAV2.1-A vector were placed symmetrically to those of the other vectors in the opposite hemisphere, and then consistent data on transgene expression levels were obtained quantitatively.

Thus, we believe that these additional experiments and data analyses augment the superiority of our mosaic vector, AAV2.1-A vector, to the other AAV vectors and underscore the utility and versatility of this vector across the animal species (for Results, see page 7, line 2 from the bottom through page 9, line 4 from the bottom; for Discussion, see page 16, lines 5-17 from the bottom). Such additional data provided in our revised manuscript greatly contribute to enhance the significance of the mosaic vector and make it more beneficial to broader readers.

2. Lastly, while it is useful to quantify expression patterns by the number of neurons or glia cells that were infected, the expression intensity seems like a poor choice (because it, as the authors acknowledge, be influenced by other factors like which cells are expressing the strongest). It would be much more meaningful to quantify the radius of expression, laminar distribution, and neuron density per injection.

We totally agree with the reviewer's criticism (see also the Responses to Major Comments 2 and 5 raised by Reviewer 2). Accordingly, counts of transgene-expressing neurons have been carried out at the sites of individual vector injections in Monkeys A-D. Consequently, strikingly similar tendencies of transgene expression were obtained between the intensity analysis (especially under the control of Syn promoter) and the cell-count analysis (see Figs. 1c,d and 3c,d; page 7, lines 2-6; page 8, lines 8-11 from the bottom).

REVIEWER COMMENTS

Reviewer #1 (Remarks to the Author):

The authors have done a very nice job responding to my comments.

Reviewer #2 (Remarks to the Author):

In the revised version of this manuscript the authors have satisfactorily addressed most of my previous comments. However, one of my major comments remains only partially answered, and I have a few other remaining comments and questions.

1. In response to my comment about the lack of assessment of anterograde and retrograde transduction after the cortical injection (comment no. 2), the authors indicated that the multiple injections performed did not allow discrimination of anterograde axonal labeling from each injection. This is understandable, but the authors should include descriptions of lack (if so) of retrograde labeled neurons in subcortical regions known to project to cortex, such as thalamus. The authors also mentioned that they did not find retrogradely labeled neurons after injections in cortex of rats. This information should be mentioned in this manuscript.

2. In response to my comment number 4, the authors indicate that they plan to report the data about the axonal transport of AAV2.1-A after injections in striatum in a future manuscript. This plan should be mentioned in the present manuscript.

3. In response to my comment number 7, the authors mentioned that no labeled neurons were found in cortical regions outside of the areas where there was leakage of the viral vector solution. This should be mentioned.

4. The authors have indicated that the phrase "sufficiently high" was deleted from the manuscript. However, there is still one instance of this phrase in line 386. The authors may consider removing or rephrasing.

5. In the Discussion the authors state that "AAV vectors without fluorescent protein tag do not likely merit application to many studies which require anatomical confirmation of vector localization/transport or histological determination of the type/number of transduced cells" (lines 382-384). I suggest revising this statement. Many non-fluorescent tag proteins are available that can be used instead of the fluorescent ones for histological purposes

Reviewer #3 (Remarks to the Author):

In this revision, the authors have attempted to address some of my previous concerns. I appreciate the addition of more animals, both NHP and rats. I also think that the quantification of effects has been improved relative to the first submission. However, I still think that the paper in its current form is not suitable for publication in Nature Communications because of limited significance. There are 2 reasons for this recommendations:

- First, the lack of repeats has only partially been addressed in the revision, and the new data actually only raise more questions. Because of how injection sites were distributed in the additional animals, the increase in repeated injection sites is very limited. Counted by animals, there are now 2 replicated injections of 2.1A with either promoter in the same brain region. To be able to treat all 3 injections in monkeys C and D as replications (either of themselves or those in monkeys A and B), some

anatomical evidence would need to be presented that they indeed fell into the same brain area – as it stands, they are spread out quite a bit along the posterior-anterior extent and more than likely were placed in different brain regions. More importantly, however, there is a huge discrepancy between the results of the first batch of animals (A, B) and the second batch (C,D), with the numbers in the second batch only reaching half of those in the first batch. This difference is not discussed in the paper at all.

- Second, the main claim of the paper is that the novel serotype can achieve better expression and neuron specificity. The newly added data show that in terms of expression levels, AAV9 is comparable (if not better) than AAV2.1-A. In terms of neuron specificity, the authors claim that AAV9 with the synapsin promoter infects more neurons than their construct. While this may be the case, Figure 4 suggests it is a rather marginal effect; more importantly, no statistics are provided to substantiate this claim. As such, the advantage of using the new construct appears very limited.

We are grateful to the three reviewers for their critical reading of our manuscript and providing us with invaluable and insightful comments. Here are point-by-point responses to the reviewers' comments.

Reviewer 1

The authors have done a very nice job responding to my comments.

We are grateful for your comments.

Reviewer 2

1. In response to my comment about the lack of assessment of anterograde and retrograde transduction after the cortical injection (comment no. 2), the authors indicated that the multiple injections performed did not allow discrimination of anterograde axonal labeling from each injection. This is understandable, but the authors should include descriptions of lack (if so) of retrograde labeled neurons in subcortical regions known to project to cortex, such as thalamus. The authors also mentioned that they did not find retrogradely labeled neurons after injections in cortex of rats. This information should be mentioned in this manuscript.

According to the reviewer's suggestion, we have closely examined the details of remote (i.e., retrograde or anterograde transneuronal) gene transduction in the thalamus after cortical injections of the AAV2.1-A vector and the other conventional serotypes of AAV vectors in the rat, and described data in the Results section of the revised manuscript (see page 10, lines 4-15; see also Fig. S1a). We have also added some descriptions that only a few thalamic neurons were transduced with the AAV2.1-A vector injected into the monkey brain (see page 10, lines 13-15; see also Fig. S1b).

2. In response to my comment number 4, the authors indicate that they plan to report the data about the axonal transport of AAV2.1-A after injections in striatum in a future manuscript. This plan should be mentioned in the present manuscript.

According to the reviewer's suggestion, we have added some descriptions of our future work to show more detailed data about the transgene expression patterns via the AAV vectors including the AAV2.1-A vector following their injections into other brain regions, such as the basal ganglia (see page 18, lines 10-14).

3. In response to my comment number 7, the authors mentioned that no labeled neurons were found in cortical regions outside of the areas where there was leakage of the viral vector solution. This should be mentioned.

In line with the reviewer's comment, we have added the information that retrograde labeling was only very rarely found in the substantia nigra (pars compacta), thalamus, or cerebral cortex, all of which are known to send projection fibers to the striatum (see page 14, lines 5-7).

4. The authors have indicated that the phrase "sufficiently high" was deleted from the manuscript. However, there is still one instance of this phrase in line 386. The authors may consider removing or rephrasing.

In line with the reviewer's comment, we have rephrased "sufficiently high" to "potentially high" in the corresponding part (see page 19, lines 5).

5. In the Discussion the authors state that "AAV vectors without fluorescent protein tag do not likely merit application to many studies which require anatomical confirmation of vector localization/transport or histological determination of the type/number of transduced cells" (lines 382-384). I suggest revising this statement. Many non-fluorescent tag proteins are available that can be used instead of the fluorescent ones for histological purposes.

In line with the reviewer's comment, we have revised the corresponding part (see page 19, lines 8-10).

Reviewer 3

First, the lack of repeats has only partially been addressed in the revision, and the new data actually only raise more questions. Because of how injection sites were distributed in the additional animals, the increase in repeated injection sites is very limited. Counted by animals, there are now 2 replicated injections of 2.1A with either promoter in the same brain region. To be able to treat all 3 injections in monkeys C and D as replications (either of themselves or those in monkeys A and B), some anatomical evidence would need to be presented that they indeed fell into the same brain area – as it stands, they are spread out quite a bit along the posterior-anterior extent and more than likely were placed in different brain regions. More importantly, however, there is a huge discrepancy between the results of the first batch of animals (A, B) and the second batch (C,D), with the numbers in the second batch only reaching half of those in the first batch. This difference is not discussed in the paper at all.

We totally agree with the reviewer's criticism that there are serious pitfalls in terms of this type of monkey research work. In the present study, we could not statistically analyze the data obtained from the monkey experiments. In our experiments for validating the properties of the AAV2.1-A vector, we used 12 different types of mKO1-expressing vectors to compare the transgene expression efficiency and neuron specificity between the AAV2.1-A vector and the other conventional serotypes of AAV vectors. In this case of quantitative study, 48 brains (at least hemispheres) with the same subjects' conditions, such as the age and gender, are in fact needed for the statistical purpose. However, considering that our best effort should be made to reduce the number of monkeys used for every research project, it was very difficult to receive the provision of individuals sufficient for statistical analysis. In our experiments, eight macaque monkeys were already used, and four of the eight monkeys (Monkeys A-D) were successfully utilized to validate the properties of the AAV2.1-A vector; two of the four monkeys were added in response to the reviewer's previous comment. Indeed, we found that injections of the AAV2.1-A vectors with CMV or synapsin promoter into eight loci of the medial frontal cortex consistently demonstrated high levels of transgene expression efficiency and neuron specificity. In our prior revision, according to the reviewer's criticism that the gene transduction properties of the vectors might differ depending on target cortical areas, the AAV2.1-A vector was injected at three loci of the medial frontal cortex, and the AAV1, 5, and 9 vectors were injected almost symmetrically on the contralateral side in Monkeys C and D. We found that although these three AAV2.1-A injections were made into somewhat distinct cortical areas, there were no marked differences in the transgene

expression patterns among them. This indicates that the properties of the AAV2.1-A vector do not substantially vary depending on target cortical areas, at least within the frontal cortex. Even if there are some differences among the injection sites of the AAV2.1-A vector, the data obtained from the injections of the other conventional serotypes of AAV vectors into the corresponding cortical areas of the opposite hemisphere clearly and consistently demonstrate the superiority of the AAV2.1-A vector. In addition, instead of additional experiments in monkeys, we have added a series of experiments in rats and performed the injections of the AAV2.1-A vector and the other conventional serotypes of AAV vectors into the same cortical areas. Based on our statistical analysis, we found that the AAV2.1-A vector exhibited a significantly higher number of transgene-expressing neurons than the AAV1, 2, and 5 vectors, and a significantly lower rate of glial infectivity than the AAV1, 5, and 9 vectors. This was exactly the same conclusion as that obtained from the monkey experiments. Therefore, we believe that the conclusion of our monkey study is sufficiently supported by these results. We have explicitly described this issue in the Results and Discussion sections (see page 9, lines 7-8; page 9, lines 1-11 from the bottom through page 14, lines 1-3; page 17, lines 8-23 through page 18, lines 11-23 from the bottom).

More importantly, however, there is a huge discrepancy between the results of the first batch of animals (A, B) and the second batch (C,D), with the numbers in the second batch only reaching half of those in the first batch. This difference is not discussed in the paper at all.

We also well understand the reviewer's criticism on the individual differences in the number of transduced neurons. As the reviewers pointed out, a lower number of transduced neurons were observed in additional experiments on Monkeys C and D than in the initial experiments on Monkeys A and B. In our study, the highest number of neurons were transduced at the site of each vector injection in Monkey A whose age was 18 years old. On the other hand, the number of transduced neurons was fewer and similar in Monkeys B-D who were relatively close in age (Monkey B, 11 years old; Monkey C, 8 years old; Monkey D, 12 years old). Although the age of subjects may have affected the transgene expression efficiency or neuron specificity of the AAV vectors, the present results were stable when the levels of transgene expression were compared in the AAV2.1 vector and the other serotype vectors which were injected in single monkeys. Furthermore, we found that higher neuron specificity (i.e., lower glial infectivity) of the AAV2.1-A

vector than that of the AAV1, 5, and 9 vectors were stable among monkeys. Honestly, it is quite difficult to minimize a bias caused by the individual differences, unlike the rodent experiments which are easy to make subjects' properties uniform, such as genetic background, gender, age, and weight. In line with the reviewer's concern, we have clearly described this issue in the Discussion section (see page 17, lines 1-9 from the bottom through page 18, lines 1-2).

Second, the main claim of the paper is that the novel serotype can achieve better expression and neuron specificity. The newly added data show that in terms of expression levels, AAV9 is comparable (if not better) than AAV2.1-A. In terms of neuron specificity, the authors claim that AAV9 with the synapsin promoter infects more neurons than their construct. While this may be the case, Figure 4 suggests it is a rather marginal effect; more importantly, no statistics are provided to substantiate this claim. As such, the advantage of using the new construct appears very limited.

We well understand the reviewer's criticism. As pointed out by the reviewer, the transgene expression level of the AAV9 vector is comparable to that of the AAV2.1-A vector. In terms of the neuron specificity, however, the AAV9 vector had a much higher level of glial infectivity than the AAV2.1-A vector in the monkey brain and, likewise, a significantly higher level of glial infectivity in the rat brain. Even though the synapsin promoter restricts the cell type to neurons in which the transgene is expressed via the vector, it is still impossible to avoid infection to non-neuronal cells (i.e., glial cells) with this serotype per se. In addition, we found that transgene expression in S100 β -positive cells remained for the AAV9 vector in the monkey brain, possibly due to leaky synapsin promoter activity or ITR promoter activity. Such viral infection to and transgene expression in glial cells may cause inflammatory changes in the brain. Previous studies (ref) have indeed reported that the AAV9 vector induces inflammatory responses due to the glial transduction. Therefore, it does not seem appropriate to use the AAV9 vector, despite such a risk, for long-term experiments, such as imaging and manipulation of neural circuits in primates. Further, the AAV2.1-A vector has already been utilized successfully in several primate studies, especially chemogenetic manipulation of neuronal activity over more than one year (for example, Nagai et al., Nature Neuroscience, 23:1157-1167, 2020), as also demonstrated in the present work. This indicates that the AAV2.1-A vector largely merits as a novel and versatile tool for systems neuroscience research. In line with the reviewer's concern, we have added some descriptions in the

Discussion section (see page 19, lines 1-11 from the bottom through page 20, lines 1-6).

REVIEWERS' COMMENTS

Reviewer #2 (Remarks to the Author):

In this new version of the manuscript the authors have addressed my remaining concerns.

Reviewer #3 (Remarks to the Author):

While my concerns regarding the variability in injection locations and expression size remain, the interpretation and caveats of the results are now more appropriately handled in the discussion, and the overall document is sufficiently improved.

We are grateful to the two reviewers for their critical reading of our manuscript and providing us with invaluable and insightful comments. Here are point-by-point responses to the reviewers' comments.

Reviewer 2

In this new version of the manuscript the authors have addressed my remaining concerns.

We greatly appreciate your comment.

Reviewer 3

While my concerns regarding the variability in injection locations and expression size remain, the interpretation and caveats of the results are now more appropriately handled in the discussion, and the overall document is sufficiently improved.

We fully understand reviewer's concerns. We sincerely appreciate your generous consideration.